# 3D Interaction Geometric Pre-training for Molecular Relational Learning

**Namkyeong Lee**[1], **Yunhak Oh**[1], **Heewoong Noh**[1], **Gyoung S. Na**[1,2],
**Minkai Xu**[3], **Hanchen Wang**[3,4] , **Tianfan Fu**[5], **Chanyoung Park**[1*]
[1] KAIST    [2] KRICT    [3] Stanford University    [4] Genentech    [5] Nanjing University

## Abstract

Molecular Relational Learning (MRL) is a rapidly growing field that focuses on understanding the interaction dynamics between molecules, which is crucial for applications ranging from catalyst engineering to drug discovery. Despite recent progress, earlier MRL approaches are limited to using only the 2D topological structure of molecules, as obtaining the 3D interaction geometry remains prohibitively expensive. This paper introduces a novel 3D geometric pre-training strategy for MRL (3DMRL) that incorporates a 3D virtual interaction environment, overcoming the limitations of costly traditional quantum mechanical calculation methods. With the constructed 3D virtual interaction environment, 3DMRL trains 2D MRL model to learn the global and local 3D geometric information of molecular interaction. Extensive experiments on various tasks using real-world datasets, including out-of-distribution and extrapolation scenarios, demonstrate the effectiveness of 3DMRL, showing up to a 24.93% improvement in performance across 40 tasks. Our code is publicly available at https://github.com/Namkyeong/3DMRL.

## 1 Introduction

Molecular relational learning (MRL) focuses on understanding the interaction dynamics between molecules and has gained significant attention from researchers thanks to its diverse applications [21, 32]. Despite recent advancements in MRL, previous works tend to ignore molecules' 3D geometric information and instead focus solely on their 2D topological structures. However, in molecular science, the 3D geometric information of molecules (Figure 1 (a)) is crucial for understanding and predicting molecular behavior across various contexts, ranging from

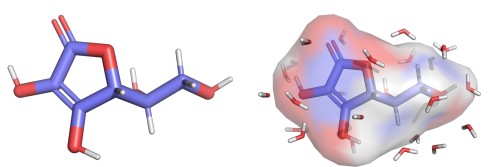

**(a) Single Molecule**      **(b) Molecular Interaction Environment**

Figure 1: 3D geometry of (a) an individual molecule and (b) the molecular interaction environment.

physical properties [1] to biological functions [10]. This is particularly important in MRL, as geometric information plays a key role in molecular interactions by determining how molecules recognize, interact, and bind with one another in their interaction environment [35]. In traditional molecular dynamics simulations, explicit solvent models, which directly consider the detailed environment of molecular interaction, have demonstrated superior performance compared to implicit solvent models, which simplify the solvent as a continuous medium, highlighting the significance of modeling the complex geometries of interaction environments [47].

However, acquiring stereochemical structures of molecules is often very costly, resulting in limited availability of such 3D geometric information for downstream tasks [24]. Consequently, in the

---

*Corresponding Author

39th Conference on Neural Information Processing Systems (NeurIPS 2025).

domain of molecular property prediction (MPP), there has been substantial progress in injecting 3D geometric information to 2D molecular graph encoders during the pre-training phase, while utilizing only the 2D molecular graph encoder for downstream tasks [36, 25]. In contrast, compared to the MPP, pre-training strategies for MRL have been surprisingly underexplored, primarily due to the following two distinct challenges in modeling complex molecular interaction environments.

Firstly, interactions between molecules occur through complex geometry as they are chaotically distributed in space as shown in Figure 1 (b). Therefore, it is essential to consider not only each molecule's independent geometry but also their relative positions and orientations in space. This requirement further complicates the acquisition of geometric information, making it more challenging to obtain detailed 3D geometry of molecular interaction environments. Consequently, it is essential to model an interaction environment that can simulate molecular interactions based solely on the 3D geometry of the individual molecules.

Secondly, even after constructing the interaction environment, how to inject the geometry between molecules during interactions are not trivial. More specifically, while the global geometry of the interaction environment is essential for understanding overall interactions and system stability, the local geometry is also critical for examining localized interactions and precise molecular behaviors. Therefore, developing pre-training strategies that effectively capture the complementary global and local geometries between molecules and their interaction environment is essential.

To address these challenges, we introduce a novel 3D geometric pre-training strategy that is applicable to various MRL models by incorporating the 3D geometry of the interaction environment for molecules (3DMRL). Specifically, instead of relying on costly traditional quantum mechanical calculation methods to obtain interaction environments, we first propose a virtual interaction environment involving multiple molecules designed to simulate real molecular interactions. Then, during the pre-training stage, a 2D MRL model is trained to produce representations that are globally aligned with those of the 3D virtual interaction environment via contrastive learning. Additionally, the 2D MRL model is trained to predict the localized relative geometry between molecules within this virtual interaction environment, allowing the model to effectively learn fine-grained atom-level interactions between molecules. These two pre-training strategies enable the 2D MRL model to be pre-trained to understand the nature of molecular interactions, facilitating positive transfer to a wide range of downstream MRL tasks. In this paper, we make the following contributions:

- Rather than relying on costly traditional quantum mechanical calculation methods to obtain interaction geometry, we propose a virtual interaction geometry made up of multiple molecules to mimic the molecular interaction environment observed in real-world conditions.

- We propose pre-training strategies that allow the 2D MRL model to learn representations of the 3D interaction environment, capturing both its global and local geometries.

- We conduct extensive experiments across various MRL models pre-trained with 3DMRL on a range of MRL tasks, including *out-of-distribution* and *extrapolation* scenarios. These experiments demonstrate improvements of up to 24.93% compared to MRL methods trained from scratch, underscoring the versatility of 3DMRL (Section 5).

To the best of our knowledge, this is the first paper proposing pre-training strategies specifically designed for molecular relational learning.

## 2 Related Works

**Molecular Relational Learning.** Molecular Relational Learning (MRL) focuses on understanding the interaction dynamics between paired molecules. Delfos [23] employs recurrent neural networks combined with attention mechanisms to predict solvation-free energy, a key factor influencing the solubility of chemical substances, using SMILES string as input. Similarly, CIGIN [32] utilizes message-passing neural networks [11] along with a cross-attention mechanism to capture atomic representations for solvation-free energy prediction. In a different context, Joung et al. [17] use graph convolutional networks [18] to generate representations of chromophores and solvents, which are then used to predict various optical and photophysical properties of chromophores, essential for developing new materials with vibrant colors. Meanwhile, MHCADDI [4] introduces a co-attentive message passing network [38] designed for predicting drug-drug interactions (DDI), which aggregates information from all atoms within a pair of molecules, not just within individual molecules. Recently,

CGIB [21] and CMRL [22] have introduced a comprehensive framework for MRL tasks, such as predicting solvation-free energy, chromophore-solute interactions, and drug-drug interactions. These models achieve this by identifying core functional groups involved in molecular interactions using information bottleneck and causal theory, respectively. However, prior studies have largely ignored molecules' 3D geometric information despite its well-established importance in comprehending various molecular properties.

**3D Pre-training for Molecular Property Prediction.** Recently, the molecular science community has shown increasing interest in pre-training machine learning models with unlabeled data, primarily due to the scarcity of labeled data for downstream tasks [22, 37, 44]. A promising approach in this area leverages molecules' inherent nature, which can be effectively represented as both 2D topological graphs and 3D geometric graphs. For instance, 3D Infomax [36] aims to enhance mutual information between 2D and 3D molecular representations using contrastive learning. GraphMVP [24] extends this concept by introducing a generative pre-training framework alongside contrastive learning. More recently, Noisy Nodes [46] and MoleculeSDE [25] have introduced methods to learn the 3D geometric distribution of molecules using a denoising framework, thereby uncovering the connection between the score function and the force field of molecules. Although the 3D structure of molecules has been effectively leveraged in pre-training for predicting single molecular properties, it remains surprisingly underexplored in the context of molecular relational learning (MRL). We provide more detailed explanations with the figure in Appendix A.

## 3  Preliminaries

### 3.1  Problem Statement

**Notations.** Given a molecule $g$, we first consider a 2D molecular graph, denoted as $g_{2D} = (\mathbf{X}, \mathbf{A})$, where $\mathbf{X} \in \mathbb{R}^{N \times F}$ represents the atom attribute matrix, and $\mathbf{A} \in \mathbb{R}^{N \times N}$ is the adjacency matrix, with $\mathbf{A}_{ij} = 1$ if a covalent bond exists between atoms $i$ and $j$. Additionally, we define a 3D conformer as $g_{3D} = (\mathbf{X}, \mathbf{R})$, where $\mathbf{R} \in \mathbb{R}^{N \times 3}$ is the matrix of 3D coordinates, each row representing the spatial position of an individual atom.

**Task Description.** Given a 2D molecular graph pair $(g_{2D}^1, g_{2D}^2)$ and 3D conformer pair $(g_{3D}^1, g_{3D}^2)$, our goal is to pre-train the 2D molecular encoders $f_{2D}^1$ and $f_{2D}^2$ simultaneously with the virtual interaction geometry $g_{vr}$, derived from the 3D conformer pair. Then, the pre-trained 2D molecular encoders $f_{2D}^1$ and $f_{2D}^2$ are utilized for various MRL downstream tasks.

### 3.2  2D MRL Model Architecture

In this paper, we mainly focus on 1) the construction of virtual interaction geometry, and 2) pre-training strategies for MRL. Therefore, we employ existing model architectures for 2D MRL, i.e., CIGIN [32], which provides a straightforward yet effective framework for MRL as depicted in Figure 2 (a). Specifically, for each pair of 2D molecular graphs, denoted as $g_{2D}^1$ and $g_{2D}^2$, the graph neural networks (GNNs)-based molecular encoders $f_{2D}^1$ and $f_{2D}^2$ initially produce an atom embedding matrix for each molecule, formulated as:

$$\mathbf{E}^1 = f_{2D}^1 \left( g_{2D}^1 \right), \quad \mathbf{E}^2 = f_{2D}^2 \left( g_{2D}^2 \right), \tag{1}$$

where $\mathbf{E}^1 \in \mathbb{R}^{N^1 \times d}$ and $\mathbf{E}^2 \in \mathbb{R}^{N^2 \times d}$ are the atom embedding matrices for $g_{2D}^1$ and $g_{2D}^2$, containing $N^1$ and $N^2$ atoms, respectively. Next, we capture the interactions between nodes in $g_{2D}^1$ and $g_{2D}^2$ using an interaction matrix $\mathbf{I} \in \mathbb{R}^{N^1 \times N^2}$, defined by $\mathbf{I}_{ij} = \text{sim}(\mathbf{E}_i^1, \mathbf{E}_j^2)$, where $\text{sim}(\cdot, \cdot)$ represents the cosine similarity measure. Subsequently, we derive new embedding matrices $\tilde{\mathbf{E}}^1 \in \mathbb{R}^{N^1 \times d}$ and $\tilde{\mathbf{E}}^2 \in \mathbb{R}^{N^2 \times d}$ for each graph, reflecting their respective interactions. This is computed using $\tilde{\mathbf{E}}^1 = \mathbf{I} \cdot \mathbf{E}^2$ and $\tilde{\mathbf{E}}^2 = \mathbf{I}^\top \cdot \mathbf{E}^1$, where $\cdot$ denotes matrix multiplication. Here, $\tilde{\mathbf{E}}^1$ represents the node embeddings of $g_{2D}^1$ that incorporates the interaction information with nodes in $g_{2D}^2$, and similarly for $\tilde{\mathbf{E}}^2$. To obtain the final node embeddings, we concatenate the original and interaction-based embeddings for each graph, resulting in $\mathbf{H}^1 = (\mathbf{E}^1 || \tilde{\mathbf{E}}^1) \in \mathbb{R}^{N^1 \times 2d}$ and $\mathbf{H}^2 = (\mathbf{E}^2 || \tilde{\mathbf{E}}^2) \in \mathbb{R}^{N^2 \times 2d}$. Finally, we apply the Set2Set function [40] to compute the graph-level embeddings $\mathbf{z}_{2D}^1$ and $\mathbf{z}_{2D}^2$ for graph $g_{2D}^1$ and $g_{2D}^2$, respectively.

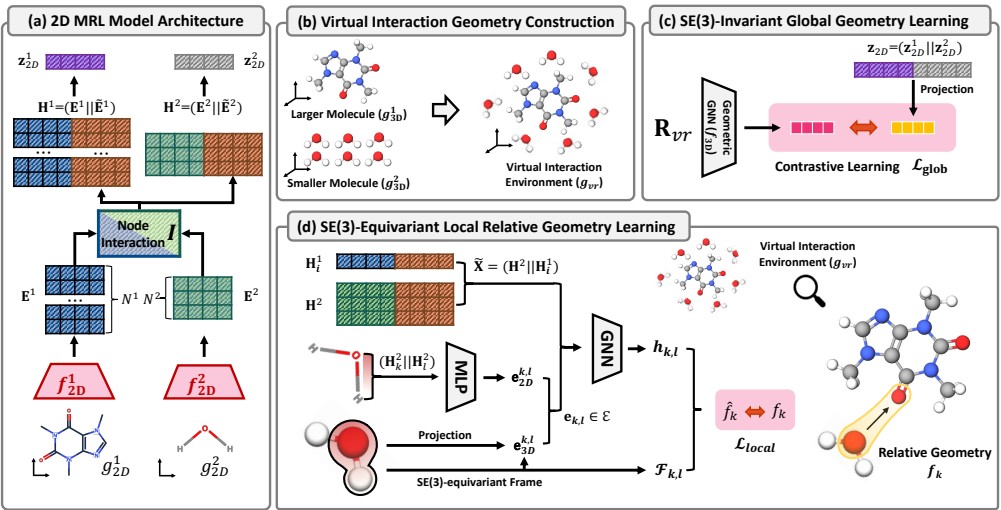

Figure 2: Overall Framework: (a) 2D MRL model architecture (Section 3.2). (b) Virtual interaction geometry construction (Section 4.1). (c) $SE(3)$-Invariant Global Geometry Learning (Section 4.2.1). (d) $SE(3)$-Equivariant Local Relative Geometry Learning (Section 4.2.2).

## 4 Methodology

In this section, we introduce our method, named 3DMRL, a novel pre-training framework for MRL utilizing 3D geometry information. Specifically, in Section 4.1, we introduce how to construct the virtual interaction geometry that can be utilized instead of expensive calculation of real interaction geometry of molecules. Then, in Section 4.2, we present two complementary geometric pre-training strategies for the 2D MRL model to acquire representations aligned with the constructed virtual interaction geometry in both global and local perspectives. The overall framework is depicted in Figure 2, and the pseudocode for the entire framework is provided in Appendix F.

### 4.1 Virtual Interaction Geometry Construction

While the 3D geometry of molecules plays a significant role in predicting molecular properties, acquiring this information involves a trade-off between cost and accuracy. For example, RDKit's ETKDG algorithm [20] is fast but less accurate. In contrast, the widely adopted metadynamics method, CREST [12], achieves a more balanced compromise between speed and accuracy, yet still requires around 6 hours to process a drug-like molecule. This challenge is even more pronounced in molecular interaction systems, which necessitates not just the geometry of individual molecules but also the relative spatial arrangements between multiple molecules [6]. Moreover, an appropriate initial geometry of a molecular interaction system is highly dependent on individual molecules in the system [27]. For this reason, a data-agnostic process for generating the initial molecular geometry is crucial for flexible and robust representation learning on molecular interaction systems.

Drawing inspiration from the explicit solvent models used in traditional molecular dynamics simulations [9], we propose a one-to-many geometric configuration that involves a relatively larger molecule $g_{3D}^1$, determined based on its radius, surrounded by multiple smaller molecules $g_{3D}^2$ as shown in Figure 2 (b) [28]. Specifically, for a given conformer pair $(g_{3D}^1 = (\mathbf{X}^1, \mathbf{R}^1), g_{3D}^2 = (\mathbf{X}^2, \mathbf{R}^2))$, we create an environment by arranging the smaller molecules $(g_{3D}^{2,1}, \ldots, g_{3D}^{2,i}, \ldots, g_{3D}^{2,n})$ around a centrally placed larger molecule $g_{3D}^1$ as follows:

**[Step 1] Select Target Atoms in the Larger Molecule.** We start by randomly selecting $n$ atoms from the larger molecule $g_{3D}^1$ that are not part of any aromatic ring. This choice is based on the fact that aromatic rings are more stable and less likely to engage in chemical reactions.

**[Step 2] Positioning the Smaller Molecules.** Each smaller molecule in $(g_{3D}^{2,1}, \ldots, g_{3D}^{2,i}, \ldots, g_{3D}^{2,n})$ is then placed close to one of the $n$ selected atoms in the larger molecule $g_{3D}^1$. This positioning

is achieved by transiting and rotating the original 3D coordinates $\mathbf{R}^2$ of the smaller molecule $g_{3D}^2$, following the method widely employed in computational chemistry [19].

- **[Step 2-1] Determine Transition Direction.** For flexible and robust molecular relational learning, we follow a widely used strategy that samples initial geometries from parameterized stochastic processes [42]. Specifically, we generate a normalized random Gaussian noise vector $\varepsilon$ (with a norm of 1), which will be used to set the direction for the transition. We then scale this direction vector $\varepsilon$ by the radius of the smaller molecule, $r^2$, to establish the transition distance.

- **[Step 2-2] Transit and Rotate to the New Position.** The new 3D coordinates for each smaller molecule are determined using the formula $\mathbf{R}^{2,i} = \mathbf{R}^2 + \varepsilon_i * r^2 + \mathbf{R}_i^1$, where $\mathbf{R}_i^1 \in \mathbb{R}^3$ represents the 3D position of the $i$-th selected atom in the larger molecule $g_{3D}^1$. This operation is performed through broadcasting, meaning $\mathbf{R}_i^1$ and $\varepsilon_i$ are added to each row of $\mathbf{R}^2$. Additionally, we apply a random rotation matrix to rotate the small molecule after its transition. This transition and rotation operations ensure that each smaller molecule is positioned close to its corresponding selected atom on the larger molecule, simulating a realistic interaction environment.

**[Step 3] Constructing Virtual Interaction Geometry.** After positioning each smaller molecule $g_{3D}^{2,i}$ near the $i$-th selected atom in the larger molecule $g_{3D}^1$, we compile all the 3D coordinates to form a unified virtual environment $g_{vr}$. This process involves combining the coordinate matrix $\mathbf{R}^1$ of the larger molecule $g_{3D}^1$, with the transited coordinates $(\mathbf{R}^{2,1}, \ldots, \mathbf{R}^{2,i}, \ldots, \mathbf{R}^{2,n})$ of the smaller molecules $(g_{3D}^{2,1}, \ldots, g_{3D}^{2,i}, \ldots, g_{3D}^{2,n})$, resulting in $\mathbf{R}_{vr} = (\mathbf{R}^1 \| \mathbf{R}^{2,1} \| \ldots \| \mathbf{R}^{2,i} \| \ldots \| \mathbf{R}^{2,n}) \in \mathbb{R}^{(N^1 + n \cdot N^2) \times 3}$. Additionally, it involves concatenating all the atom attribute matrices to form $\mathbf{X}_{vr} = (\mathbf{X}^1 \| \mathbf{X}^2 \| \ldots \| \mathbf{X}^2) \in \mathbb{R}^{(N^1 + n \cdot N^2) \times F}$, thereby defining the virtual interaction geometry as $g_{vr} = (\mathbf{X}_{vr}, \mathbf{R}_{vr})$. Note that multiple small molecules share the same attribute matrix $\mathbf{X}^2$, since we use the atom attribute irrelevant to the atomic coordinates.

Note that such randomized configurations of interaction environment is a well-established strategy in molecular simulations. For instance, protein–ligand docking protocols (e.g., Rosetta) often initialize ligands in random orientations relative to the protein before searching for binding modes. Similarly, Monte Carlo insertion methods like Widom's test-particle approach randomly insert solvent molecules to explore solute–solvent configurations without bias. Moreover, while we construct the virtual interaction geometry (**Step 1** to **Step 3**) at each epoch during the pre-training phase, the virtual environment can be generated in real time because transition and rotation are matrix operations. Therefore, we argue that our approach allows efficient sampling over a wide range of distances and orientations while remaining physically sound: in the limit of sufficient sampling, no unphysical configuration is favored, and the process mimics the early stages of solvation when solvent molecules approach from arbitrary directions. In Section 5 and Appendix E.4, we analyze the environment in various aspects, justifying the proposed approach for constructing a virtual interaction environment.

## 4.2 Pre-training Strategies

Once the virtual interaction geometry is established, we pre-train the 2D MRL model using two complementary geometry learning strategies: $SE(3)$-invariant global geometry learning (Section 4.2.1) and $SE(3)$-equivariant local relative geometry learning (Section 4.2.2).

### 4.2.1 $SE(3)$-**Invariant Global Geometry Learning**

Given a paired 2D molecular graphs $(g_{2D}^1, g_{2D}^2)$ and its corresponding 3D virtual interaction geometry $g_{vr}$, we first encode them with a 2D MRL model, and a geometric deep learning model, respectively. For 2D molecular graphs, we compute the molecule-level representations, $\mathbf{z}_{2D}^1$ and $\mathbf{z}_{2D}^2$, for each molecule $g_{2D}^1$ and $g_{2D}^2$, respectively, as outlined in the Section 3.2. Following this, we derive the 2D interaction representation $\mathbf{z}_{2D}$, by concatenating these two representations, i.e., $\mathbf{z}_{2D} = (\mathbf{z}_{2D}^1 \| \mathbf{z}_{2D}^2)$. On the other hand, to encode the 3D virtual interaction geometry $g_{vr} = (\mathbf{X}_{vr}, \mathbf{R}_{vr})$, we use geometric GNNs $f_{3D}$ that output $SE(3)$ invariant [7] representations $\mathbf{z}_{3D}$ given the coordinates of atoms $\mathbf{R}_{vr}$ in virtual interaction geometry [34], i.e., $\mathbf{z}_{3D} = f_{3D}(\mathbf{R}_{vr})$. Then, as shown in Figure 2 (c), we align the 2D interaction representation $\mathbf{z}_{2D}$ and the 3D geometry representation $\mathbf{z}_{3D}$ via Normalized

temperature-scaled cross entropy loss [3] as follows:

$$\mathcal{L}_{\text{glob}} = -\frac{1}{N_{\text{batch}}} \sum_{i=1}^{N_{\text{batch}}} \left[ \log \frac{e^{\text{sim}(\mathbf{z}_{2\text{D},i}, \mathbf{z}_{3\text{D},i})/\tau}}{\sum_{k=1}^{N_{\text{batch}}} e^{\text{sim}(\mathbf{z}_{2\text{D},i}, \mathbf{z}_{3\text{D},k})/\tau}} + \log \frac{e^{\text{sim}(\mathbf{z}_{3\text{D},i}, \mathbf{z}_{2\text{D},i})/\tau}}{\sum_{k=1}^{N_{\text{batch}}} e^{\text{sim}(\mathbf{z}_{3\text{D},i}, \mathbf{z}_{2\text{D},k})/\tau}} \right].$$

where $\text{sim}(\cdot, \cdot)$ represents cosine similarity, $\tau$ denotes the temperature hyperparameter, and $N_{\text{batch}}$ refers to the number of pairs within a batch. By training the 2D MRL model to output interaction representations that align with the 3D interaction geometry, the model effectively learns the overall global geometry of molecular interactions during the pre-training phase.

### 4.2.2   $SE(3)$-Equivariant Local Relative Geometry Learning

Beyond the overall global geometry of interaction, it is essential to learn about the intermolecular local relative geometry between molecules during molecular interactions, as their localized relative geometry governs how molecules interact in various environments. To achieve this, we propose pre-training the 2D MRL model to learn local relative geometry by predicting the 3D geometry of the paired molecule, specifically by training a smaller 2D molecule encoder to predict the geometry of the larger molecule. However, predicting relative geometry from a 2D representation is challenging because the prediction must adhere to the physical properties of the molecule, specifically being equivariant to rotations and transitions in 3D Euclidean space, also known as *SE(3)-equivariance* [7]. To address this, we propose predicting the relative geometry between molecules by utilizing local frame [5], which allows for flexible conversion between invariant and equivariant features.

More specifically, given the position $\mathbf{R}^{2,i}$ of the $i$-th small molecule $g_{3\text{D}}^{2,i}$ in the constructed virtual interaction geometry, we first define an orthogonal local frame $\mathcal{F}_{k,l}$ between atoms $k$ and $l$ within molecule $g_{3\text{D}}^{2,i}$ as follows:

$$\mathcal{F}_{k,l} = \left( \frac{\mathbf{r}_k - \mathbf{r}_l}{||\mathbf{r}_k - \mathbf{r}_l||}, \frac{\mathbf{r}_k \times \mathbf{r}_l}{||\mathbf{r}_k \times \mathbf{r}_l||}, \frac{\mathbf{r}_k - \mathbf{r}_l}{||\mathbf{r}_k - \mathbf{r}_l||} \times \frac{\mathbf{r}_k \times \mathbf{r}_l}{||\mathbf{r}_k \times \mathbf{r}_l||} \right), \tag{2}$$

where $\mathbf{r}_k \in \mathbb{R}^3$ and $\mathbf{r}_l \in \mathbb{R}^3$ indicate the position of atoms $k$ and $l$ in constructed virtual interaction geometry, respectively. For simplicity, please note that we will omit the molecule index $i$ in the notation from here. With the established local frame, we derive the invariant 3D feature for the edge between atoms $k$ and $l$ by projecting their coordinates into the local frame, i.e., $\mathbf{e}_{3\text{D}}^{k,l} = \text{Projection}_{\mathcal{F}_{k,l}}(\mathbf{r}_k, \mathbf{r}_l) \in \mathbb{R}^d$. Additionally, we obtain the 2D invariant edge feature between atoms $k$ and $l$ by concatenating the respective features from the 2D molecular graph, i.e., $\mathbf{e}_{2\text{D}}^{k,l} = \text{MLP}(\mathbf{H}_k^2 || \mathbf{H}_l^2) \in \mathbb{R}^d$. Now that we have both invariant 2D and 3D features, we can derive the final invariant edge feature $\mathbf{e}^{k,l}$ by combining these invariant edge features as follows:

$$\mathbf{e}_{k,l} = \mathbf{e}_{2\text{D}}^{k,l} + \mathbf{e}_{3\text{D}}^{k,l}. \tag{3}$$

We define the edge feature set $\mathcal{E}$, which includes $\mathbf{e}_{k,l}$ for every possible pair of atoms.

With the invariant final edge feature set $\mathcal{E}$, we can further process the small molecule information through GNNs to predict the geometry of the larger molecule. To achieve this, we first obtain the atom features specific to the $i$-th small molecule by concatenating the $i$-th atom representation of the larger molecule (to which the $i$-th small molecule is assigned) with each atom representation of the small molecule, i.e., $\tilde{\mathbf{X}} = (\mathbf{H}^2 || \mathbf{H}_i^1) \in \mathbb{R}^{N^2 \times 4d}$ using broadcasting. This approach allows the model to learn a more precise geometry by incorporating the features of the assigned atom in the larger molecule. Next, with the edge feature set $\mathcal{E}$ and the atom feature $\tilde{\mathbf{X}}$, we derive the final edge representation $\mathbf{h}_{k,l}$ through multiple GNN layers, represented as $\mathbf{h}_{k,l} = \text{GNN}(\tilde{\mathbf{X}}, \mathcal{E})$. Finally, we determine the relative geometry $\hat{f}_k$ between the atom $k$ of the small molecule and the central larger molecule by combining the final invariant edge representation $\mathbf{h}_{k,l}$ with our $SE(3)$-equivariant frame $\mathcal{F}_{k,l}$ as follows:

$$\hat{f}_k = \sum_l \mathbf{h}_{k,l} \odot \mathcal{F}_{k,l}, \tag{4}$$

where $\odot$ indicates element-wise product. This approach guarantees our predicted relative geometry $\hat{f}_k$ to be $SE(3)$-equivariant. Then, we calculate the relative geometry prediction loss as follows:

$$\mathcal{L}_{\text{local}} = \frac{1}{n \cdot N^2} \sum_{i=1}^{n} \sum_{k=1}^{N^2} ||f_k^i - \hat{f}_k^i||_2^2, \tag{5}$$

where $f_k^i$ represents the ground truth relative geometry between the larger molecule and the $k$-th atom of the $i$-th small molecule. We define the relative geometry $f_k^i$ as the direction between the $k$-th atom of the $i$-th small molecule and the $i$-th atom of the larger molecule to which the small molecule is attached, i.e., $f_k^i = (\mathbf{R}_k^{2,i} - \mathbf{R}_i^1)/||\mathbf{R}_k^{2,i} - \mathbf{R}_i^1||_2$. Note that $\mathcal{L}_{\text{local}}$ is calculated for every molecule pair in the batch, although we have omitted this notation for simplicity.

Finally, we pre-train the 2D MRL model by jointly optimizing two proposed losses, i.e., $SE(3)$-invariant global geometry loss and $SE(3)$-equivariant local relative geometry loss, as follows:

$$\mathcal{L}_{\text{pre-train}} = \mathcal{L}_{\text{glob}} + \alpha \cdot \mathcal{L}_{\text{local}}, \qquad (6)$$

where $\alpha$ is a hyperparameter that determines the trade-off between the global geometry loss and the local geometry loss. After task-agnostic pre-training, the 2D molecular encoders $f_{\text{2D}}^1$ and $f_{\text{2D}}^2$ are fine-tuned for specific downstream tasks where access to 3D geometric information is limited.

### 4.3 Discussion

While we define local relative geometry for learning fine-grained interactions between molecules, we can view local relative geometry as an *interaction force* between molecules. This provides a physically motivated supervision signal rooted in classical intermolecular forces, many of which are central and act along the internuclear axis. For example, van der Waals interactions (described by the Lennard-Jones potential) exhibit repulsive or attractive forces directed along this axis. At short distances, repulsion dominates and aligns directly outward between nuclei.

This supervision scheme serves as a central-force approximation, consistent with classical force fields, and offers a lightweight surrogate for full force labels, which would require costly quantum chemistry or MD simulations. Notably, SchNet[34] demonstrated that even approximate force signals improve learning of molecular interactions. Our direction-based supervision enables the model to learn geometric features like hydrogen bond alignment or steric repulsion trajectories in an SE(3)-equivariant manner.

Since solvent atoms are placed near specific solute atoms, the dominant interaction direction aligns with the interatomic vector, making it a reasonable proxy for the net force axis. Thus, the unit direction vector serves as a pseudo-force label, conveying the primary interaction axis and encouraging the model to encode directionality of intermolecular interactions.

## 5 Experiments

### 5.1 Experimental Setup

**Downstream Tasks.** Following a prior study [21], we employ **ten** datasets to comprehensively evaluate the performance of 3DMRL on two tasks: 1) molecular interaction prediction, and 2) drug-drug interaction (DDI) prediction. For the molecular interaction prediction task, we utilize the **Chromophore** dataset [16], which pertains to three optical properties of chromophores, along with five other datasets related to the solvation free energy of solutes: **MNSol** [26], **FreeSolv** [29], **CompSol** [30], **Abraham** [13], and **CombiSolv** [39]. In the Chromophore dataset, we focus on the maximum absorption wavelength (**Absorption**), maximum emission wavelength (**Emission**), and excited state lifetime (**Lifetime**) properties. For the DDI prediction task, we use two datasets: **ZhangDDI** [48] and **ChChMiner** [49], both of which contain labeled DDI data. We provide further details on pre-training and downstream task datasets in Appendix B.1 and B.2, respectively.

**Baseline methods.** We validate the effectiveness of 3DMRL by using it to enhance various recent state-of-the-art molecular relational learning methods, including **MPNN** [11], **AttentiveFP** [43], **CIGIN** [32], **CGIB** [21], and **CGIB$_{\text{Cont}}$** [21]. Additionally, we compare our proposed pre-training framework, 3DMRL, with recent molecular pre-training approaches that aim to learn 3D structure of individual molecules, such as **3D Infomax** [36], **GraphMVP** [24], and **MoleculeSDE** [25]. It is important to note that these approaches involve pre-training a single encoder for molecular property prediction (**MPP Pre-training** in Table 2), whereas our work is pioneering in training two separate encoders simultaneously during pre-training for molecular relational learning (**MRL Pre-training** in Table 2). For the baseline methods, we use the original authors' code and conduct the experiments in the same environment as 3DMRL to ensure a fair comparison. Moreover, while we choose to mainly

Table 1: Performance improvement in molecular interaction tasks across different models with our proposed pre-training strategy (RMSE) (↓). We conduct 15 independent runs for each model and report their mean along with the standard deviation (in parentheses). Colors indicate the performance improvement compared to the models trained from scratch.

| Model | Chromophore | | | MNSol | FreeSolv | CompSol | Abraham | CombiSolv |
|---|---|---|---|---|---|---|---|---|
| | Absorption | Emission | Lifetime | | | | | |
| MPNN | 22.00 (0.30) | 26.34 (0.41) | 0.789 (0.021) | 0.643 (0.005) | 1.127 (0.110) | 0.420 (0.018) | 0.640 (0.008) | 0.614 (0.031) |
| + 3DMRL | 19.96 (0.12) | 25.21 (0.31) | 0.753 (0.018) | 0.609 (0.008) | 1.068 (0.087) | 0.377 (0.020) | 0.550 (0.051) | 0.599 (0.025) |
| Improvement | 9.27% | 4.29% | 4.56% | 5.28% | 5.24% | 10.24% | 14.06% | 2.44% |
| AttentiveFP | 22.86 (0.30) | 28.70 (0.23) | 0.871 (0.010) | 0.570 (0.021) | 1.019 (0.070) | 0.350 (0.008) | 0.426 (0.042) | 0.471 (0.028) |
| + 3DMRL | 22.80 (0.61) | 28.54 (1.97) | 0.784 (0.013) | 0.562 (0.031) | 0.901 (0.059) | 0.271 (0.009) | 0.378 (0.027) | 0.448 (0.011) |
| Improvement | 0.26% | 0.55% | 9.99% | 1.40% | 11.57% | 22.57% | 11.26% | 4.88% |
| CIGIN | 19.66 (0.69) | 25.84 (0.23) | 0.821 (0.017) | 0.582 (0.022) | 0.958 (0.116) | 0.369 (0.018) | 0.421 (0.018) | 0.464 (0.002) |
| + 3DMRL | 18.00 (0.17) | 24.21 (0.09) | 0.729 (0.014) | 0.528 (0.019) | 0.839 (0.105) | 0.277 (0.006) | 0.371 (0.031) | 0.435 (0.006) |
| Improvement | 8.44% | 6.30% | 11.20% | 9.28% | 12.42% | 24.93% | 11.87% | 6.25% |
| CGIB | 18.37 (0.35) | 24.52 (0.25) | 0.808 (0.015) | 0.562 (0.008) | 0.876 (0.037) | 0.321 (0.002) | 0.404 (0.037) | 0.448 (0.008) |
| + 3DMRL | 17.93 (0.35) | 23.92 (0.29) | 0.733 (0.009) | 0.538 (0.020) | 0.842 (0.046) | 0.274 (0.002) | 0.370 (0.027) | 0.442 (0.015) |
| Improvement | 2.40% | 5.90% | 9.28% | 4.27% | 3.88% | 14.64% | 8.42% | 1.33% |
| CGIB$_{Cont}$ | 18.59 (0.24) | 24.68 (0.49) | 0.803 (0.019) | 0.561 (0.012) | 0.897 (0.098) | 0.333 (0.005) | 0.404 (0.039) | 0.452 (0.015) |
| + 3DMRL | 17.90 (0.17)** | 23.94 (0.24) | 0.720 (0.020) | 0.524 (0.018)* | 0.863 (0.075) | 0.284 (0.007) | 0.372 (0.021) | 0.441 (0.022) |
| Improvement | 3.71% | 3.00% | 10.33% | 6.59% | 3.79% | 14.71% | 7.92% | 2.43% |

compare 2D encoder pre-training approach, we also compare 3D encoder pre-training approaches [15, 31, 8] in Appendix E.5. We provide more details on the compared methods in Appendix C.

**Evaluation protocol.** Following Pathak et al. [32], for the molecular interaction prediction task, we evaluate the models under a 5-fold cross-validation scheme. The dataset is randomly split into 5 subsets and one of the subsets is used as the test set, while the remaining subsets are used to train the model. A subset of the test set is selected as the validation set for hyperparameter selection and early stopping. We repeat 5-fold cross-validation three times (i.e., 15 runs in total) and report the accuracy and standard deviation of the repeats. For the DDI prediction task [21], we conduct experiments on two different *out-of-distribution* scenarios, namely **molecule split** and **scaffold split**. For the **molecule split**, the performance is evaluated when the models are presented with new molecules not included in the training dataset. In the **scaffold split** setting [14], just like in the molecule split, molecules corresponding to scaffolds that were not seen during training will be used for testing. For both splits, we repeat 5 independent experiments with different random seeds on split data, and report the accuracy and the standard deviation of the repeats. In both scenarios, we split the data into training, validation, and test sets with a ratio of 60/20/20%. We provide details on evaluation protocol, model implementation, and model training in Section D.

## 5.2 Experimental Results

We begin by comparing each model architecture trained from scratch with the same architecture pre-trained using our proposed strategy, referred to as +3DMRL in Table 1. We have the following observations: **1)** 3DMRL obtains consistent improvements over the base graph neural networks in all 40 tasks (across various datasets and neural architectures), achieving up to 24.93% relative reduction in RMSE. While the paper is written based on CIGIN for better understanding in Section 3.2, we could observe performance improvements not only in CIGIN but also in various other model architectures, demonstrating the versatility of proposed pre-training strategies. We further demonstrate how our pre-training strategies are adopted to various model architectures in Appendix C.

Additionally, we compare our pre-training strategies with recent molecular pre-training approaches proposed for molecular property prediction (MPP) of a single molecule. Table 2 (a) and (b) show the results for the molecular interaction prediction task, and the drug-drug interaction (DDI) task, respectively. As these approaches are originally designed for single molecules, we first pre-train the GNNs using each strategy, then incorporate the pre-trained GNNs into the CIGIN architecture and fine-tune them for various MRL downstream tasks. We have the following observations: **2)** Although MPP pre-training methods have demonstrated success in molecular property prediction in prior studies, they did not yield satisfactory results in molecular relational learning tasks and, in some cases, even resulted in negative transfer. This highlights the need for creating specialized pre-training strategies tailored to MRL tasks. We further demonstrate the MPP pre-training strategy with a large-scale dataset still performs worse than 3DMRL in Appendix E.1. **3)** On the other hand, pre-training

Table 2: Performance of CIGIN model on (a) molecular interaction tasks using different pre-training strategies (RMSE) (↓) and (b) out-of-distribution DDI tasks using different pre-training strategies (AUROC) (↑). For each dataset, we highlight the best method **in bold**.

| Strategy | (a) Molecular Interaction Tasks (RMSE ↓) | | | | | | | | (b) Drug-Drug Interaction Task (AUROC ↑) | | | |
| | Chromophore | | | MNSol | FreeSolv | CompSol | Abraham | CombiSolv | (c) Molecule Split | | (d) Scaffold Split | |
| | Absorption | Emission | Lifetime | | | | | | ZhangDDI | ChChMiner | ZhangDDI | ChChMiner |
|---|---|---|---|---|---|---|---|---|---|---|---|---|
| No Pre-training | 19.66 (0.69) | 25.84 (0.23) | 0.821 (0.017) | 0.567 (0.014) | 0.884 (0.074) | 0.331 (0.029) | 0.412 (0.028) | 0.458 (0.002) | 71.75 (0.76) | 76.21 (1.19) | 70.96 (1.40) | 75.81 (0.79) |
| **MPP (molecular property prediction) Pre-training** | | | | | | | | | | | | |
| 3D Infomax | 18.71 (0.61) | 24.59 (0.22) | 0.790 (0.022) | 0.585 (0.015) | 0.873 (0.103) | 0.321 (0.041) | 0.426 (0.036) | 0.464 (0.004) | 71.01 (2.19) | 76.05 (1.30) | 70.90 (1.63) | 74.87 (1.08) |
| GraphMVP | 18.40 (0.62) | 24.73 (0.14) | 0.797 (0.022) | 0.561 (0.025) | 1.010 (0.115) | 0.301 (0.025) | 0.418 (0.020) | 0.437 (0.015) | 71.82 (1.44) | 76.42 (1.68) | 71.73 (0.95) | 76.13 (1.01) |
| MoleculeSDE | 18.56 (0.24) | 24.91 (0.10) | 0.836 (0.040) | 0.564 (0.018) | 0.971 (0.122) | 0.308 (0.024) | 0.426 (0.028) | 0.454 (0.012) | 70.07 (0.58) | 76.37 (1.14) | 69.46 (1.55) | 76.03 (1.13) |
| **MRL (molecular relational learning) Pre-training** | | | | | | | | | | | | |
| 3DMRL | **18.00** (0.17) | **24.21** (0.09) | **0.729** (0.014) | **0.528** (0.019) | **0.839** (0.105) | **0.277** (0.006) | **0.371** (0.031) | **0.435** (0.006) | **74.00** (0.72) | **78.93** (0.59) | **74.85** (1.58) | **78.56** (1.03) |

with 3DMRL consistently delivers significant performance improvements across downstream tasks. This validates the effectiveness of our approach, as it successfully integrates scientific knowledge into the pre-training strategy, enhancing the model's overall performance. **4)** Additionally, for the DDI task in Table 2 (b), we observed that the performance improvement is more pronounced in challenging scenarios ((d) Scaffold split) compared to less difficult ones ((c) Molecule split). This highlights the enhanced generalization ability of 3DMRL in out-of-distribution scenarios, demonstrating its potential for drug discovery applications where robust generalization across unknown molecules is essential. We explore the *extrapolation* capability of 3DMRL in Appendix E.2.

## 5.3 Model Analysis

**Ablation Studies.** To further understand our model, we conduct an ablation study to investigate the impact of two key components on the final performance. Specifically, as shown in Equation 6, the objective function contains two terms: (i) global geometry loss and (ii) intermolecular local geometry loss; we curate two variants that involve only (i) (denoted **only glob.**) and only (ii) (denoted **only local**) in Figure 3. As shown in Figure 3, learning the global geometry plays a particularly critical role. Removing it from 3DMRL results in a significant performance drop, even falling below MPP pre-training strategies such as 3D Infomax and GraphMVP. This is because the global

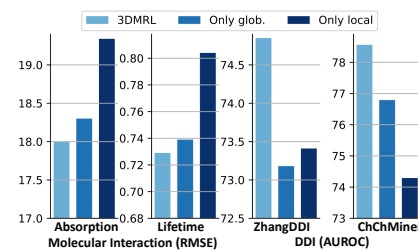

Figure 3: Ablation studies.

geometry loss allows the model to capture the overall interaction geometry at the molecular level, while the local geometry loss focuses on learning more fine-grained, atom-level interactions. However, combining both losses, as in 3DMRL, yields the best results, demonstrating the importance of leveraging the strengths of both levels of granularity. We provide further detailed results of ablation studies in Appendix E.3.

**Molecule Collision Analysis.** While the virtual environment is designed to carefully mimic the nature of molecular interactions, as discussed in Section 4.1, molecule collisions can still occur within the environment. To first examine how molecule collisions affect model performance, we created a "No Radius" model that does not take the radius into account during pre-training. Looking at the atomic overlap in Table 3, we observed that 3DMRL, which utilizes radius in-

Table 3: Model performance in various 3D interaction environments with reduced collision.

| | Atomic Overlap | Time (min/epoch) | Performance | | |
| | | | Absorption | Emission | Lifetime |
|---|---|---|---|---|---|
| No Radius | 73.84 % | 5.30 | 18.68 | 25.37 | 0.745 |
| Fixed Direction | 19.28 % | 5.30 | 18.26 | 24.24 | 0.734 |
| Twice Radius | 10.28 % | 5.32 | 18.23 | 24.25 | 0.730 |
| Regenerate | 0.0% | 23.01 | 18.20 | **23.86** | **0.727** |
| 3DMRL | 25.12% | 5.32 | **18.00** | 24.21 | 0.729 |

formation, significantly reduces the overlap ratio between molecules compared to the "No Radius" configuration. Moreover, we found that in the "No Radius" case, where the atomic overlap ratio is very high, the performance was much lower than that of the 3DMRL. To further investigate whether further reducing atomic overlap would be helpful, we experimented with several additional configurations. The "Fixed direction" configuration was designed to prevent overlap caused by random direction placement by positioning the solvent along the direction from the origin to the target atom. "Twice radius" refers to multiplying the radius in the equation in line 170. These methods reduce atomic overlap by decreasing randomness and increasing the distance between molecules, respectively; however, in terms of performance, they were either similar to or worse than 3DMRL.

The experimental results showed that both methods were able to reduce atomic overlap, but in terms of performance, they were either similar to or worse than 3DMRL. Lastly, we introduce "Regenerate," which regenerates the 3D virtual environment every time a collision occurs between any molecules in a virtual environment. Although the collision between molecules can certainly be avoided in this case, this approach incurs high computational complexity. In Table 3, we observe that the performance gain of "Regenerate" is minimal, despite its significantly higher computational requirements. Based on these results, we argue that 3DMRL strikes an appropriate balance between computation and performance.

**Sensitivity analysis on $n$.** Moreover, we conduct a sensitivity analysis to explore the empirical effect of the number of target atoms $n$, which determines the number of small molecules in a virtual interaction geometry. In Figure 4 (a), we observe that the model performs the best when using five small molecules to construct the virtual interaction geometry. More specifically, using too few small molecules ($n = 2$) results in poorer performance, as it fails to adequately simulate real-world interaction environments. On the other hand, the model performance also declines as the number of small molecules increases, likely due to the 3D geometry encoder overfitting to the small

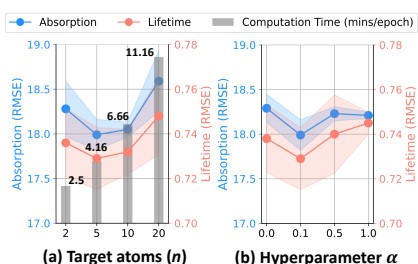

Figure 4: Sensitivity analysis.

molecules with an excessive count. Furthermore, we observe that as the number of target atoms increases, more extensive computational resources are required to encode the 3D interaction geometry during pre-training. Hence, selecting an appropriate number of target atoms is crucial for both model performance and computational efficiency. We provide further analyses on virtual interaction geometry and other tasks in Appendix E.4.

**Sensitivity analysis on $\alpha$.** We conduct sensitivity analysis on $\alpha$, which controls the weight of local geometry loss, in Equation 6. In Figure 4 (b), the model's performance declines as $\alpha$ increases from 0.1, primarily because it overly emphasizes atom-level interactions between the molecules instead of considering the overall interaction geometry. Conversely, we also notice a drop in performance when local geometry loss is not utilized ($\alpha = 0.0$), as this causes the model to lose ability in learning fine-grained atom-level interactions. It is important to note that while we set $n = 5$ and $\alpha = 0.1$ during pre-training, models pre-trained with varying $n$ and $\alpha$ consistently outperform those trained from scratch, demonstrating 3DMRL's robustness.

# 6   Conclusion

In this work, we propose 3DMRL, a novel pre-training framework that effectively integrates 3D geometric information into MRL. By constructing a virtual interaction geometry and utilizing local and global geometry prediction, our approach effectively incorporates complex 3D interaction geometry information into 2D MRL models. Experimental results demonstrate that 3DMRL significantly enhances the performance of 2D MRL models across various downstream tasks and neural architectures, validating the importance of incorporating 3D geometric data.

For **future work**, we intend to develop and train a virtual interaction geometry generator capable of mimicking MD trajectories of molecular interactions. We will then substitute this generator for the purely random generation method currently used in Section 4.1, providing a more physically informed signal.. Furthermore, we plan to broaden this research in drug-target binding affinity prediction, a core task in drug discovery that involves complex protein structures as the larger molecule.

# Acknowledgement

This work was supported by the Institute of Information & Communications Technology Planning & Evaluation (IITP) grant funded by the Korea government (MSIT) (RS-2025-02304967, AI Star Fellowship (KAIST)). Additionally, this research received funding from the National Research Foundation of Korea (NRF) through two separate grants: RS-2024-00335098 (funded by the Korea government (MSIT)) and RS-2022-NR068758 (funded by the Ministry of Science and ICT).

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

# Supplementary Material for
# 3D Interaction Geometric Pre-training for Molecular Relational Learning

# A Molecular Relational Learning

In this section, we provide further clarification on molecular relational learning by contrasting it with conventional molecular property prediction tasks. As illustrated in Figure 5 (a), conventional molecular property prediction focuses on learning the properties of a single molecule. Models like GraphMVP, 3D Infomax, and MoleculeSDE utilize the 3D information of individual molecules during pre-training to improve performance in downstream tasks aimed at predicting single molecular properties.

In contrast, as shown in Figure 5 (b), molecular relational learning focuses on learning the properties of molecules after their interactions. Our pre-training approach trains both encoders simultaneously to learn 3D information from the virtual environment $g_{vr}$. What sets our approach apart from traditional molecular pretraining is its specific tailoring to our Molecular Relational Learning strategy. This allows the two encoders to learn how paired molecules interact in 3D space, which is essential for various downstream tasks in Molecular Relational Learning.

Figure 5: Difference between the conventional pre-training strategy for (a) molecular property prediction and our (b) molecular relational learning.

# B Datasets

## B.1 Pre-Training Datasets

We utilize three distinct datasets, i.e., **Chromophore**, **CombiSolv**, and **DDI**, to pre-train 3DMRL for each downstream task as described in Section 5. Specifically, we use the **Chromophore** dataset for downstream tasks involving the optical properties of chromophores, the **CombiSolv** dataset for tasks related to the solvation free energy of solutes, and the **DDI** dataset, which we created for the drug-drug interaction task.

- The **Chromophore** dataset [16] consists of 20,236 combinations derived from 6,815 chromophores and 1,336 solvents, provided in SMILES string format. For pre-training, we initially convert chromophores and solvents into their respective 3D structures via rdkit, resulting in 6,524 3D structures for chromophores and 1,255 for solvents. These 6,524 unique chromophores are then randomly paired with the 1,255 solvents to generate a sufficient number of pairs. Out of the possible 8,187,620 chromophore-solvent combinations, we randomly sample 1%, which corresponds to 81,876 pairs, for pre-training.

- The **CombiSolv** dataset [39] contains 10,145 combinations derived from 1,368 solutes and 291 solvents, provided in SMILES string format. Similar to our approach with the Chromophore dataset, we first convert solutes and solvents into their corresponding 3D structures, yielding

1,368 3D structures for solutes and 290 for solvents. From the potential random combinations, we select 79,344 solute-solvent pairs, representing 20% of all possible pairs.

- For the **DDI** dataset, we compile drug-drug pairs from the ZhangDDI [48], ChChMiner [49], and DeepDDI [33] datasets. From a total of 235,547 positive pairs, we randomly sample 40% (i.e., 94,218 pairs) for use as the pre-training dataset. While chromophores and solutes act as the larger molecule $g^1$ in molecular interaction tasks, in the DDI dataset, we designate the drug with the larger radius as the larger molecule.

## B.2 Downstream Task Datasets

**Molecular Interaction Prediction.** For the molecular interaction prediction task, we transform the SMILES strings into graph structures using the CIGIN implementation available on GitHub [2][32]. Regarding the datasets related to solvation free energies, such as MNSol, FreeSolv, CompSol, Abraham, and CombiSolv, we utilize SMILES-based datasets from previous studies [39]. Following previous work [21], we specifically filter the data to include only solvation free energies measured at temperatures of 298 K ($\pm$ 2) and exclude any data involving ionic liquids and ionic solutes [39].

- The **Chromophore** dataset [16] consists of 20,236 combinations derived from 6,815 chromophores and 1,336 solvents, provided in SMILES string format. This dataset includes optical properties sourced from scientific publications, with unreliable experimental results being excluded after thorough examination of absorption and emission spectra. In our work, we assess model performance by predicting three key properties: **maximum absorption wavelength (Absorption)**, **maximum emission wavelength (Emission)**, and **excited state lifetime (Lifetime)**, which are crucial for designing chromophores for specific applications. To ensure the integrity of each dataset, we remove any NaN values that were not reported in the original publications. Additionally, following previous work [21], for the Lifetime data, we apply log normalization to the target values to mitigate skewness in the dataset, thereby enhancing training stability.

- The **MNSol** dataset [26] features 3,037 experimentally measured free energies of solvation or transfer for 790 distinct solutes and 92 solvents. For our study, we focus on 2,275 pairs comprising 372 unique solutes and 86 solvents, in alignment with prior research [39].

- The **FreeSolv** dataset [29] offers 643 hydration free energy values, both experimental and calculated, for small molecules in water. In our research, we utilize 560 experimental measurements, consistent with the dataset selection criteria from previous studies [39].

- The **CompSol** dataset [30] has been designed to illustrate the impact of hydrogen-bonding association effects on solvation energies. For our study, we analyze 3,548 solute-solvent pairs, encompassing 442 distinct solutes and 259 solvents, in accordance with prior research parameters [39].

- The **Abraham** dataset [13], curated by the Abraham research group at University College London, provides extensive data on solvation. For this study, we focus on 6,091 solute-solvent combinations, comprising 1,038 distinct solutes and 122 solvents, as outlined in previous research [39].

- The **CombiSolv** dataset [39] integrates the data from MNSol, FreeSolv, CompSol, and Abraham, encompassing a total of 10,145 solute-solvent combinations. This dataset features 1,368 unique solutes and 291 distinct solvents.

**Drug-Drug Interaction (DDI) Prediction.** In the drug-drug interaction prediction task, we utilize the positive drug pairs provided in the MIRACLE GitHub repository[3], which excludes data instances that cannot be represented as graphs from SMILES strings. To create negative samples, we generate a corresponding set by sampling from the complement of the positive drug pairs. This approach is applied to both datasets. Additionally, for the classification task, we adhere to the graph conversion process outlined by MIRACLE [41].

- The **ZhangDDI** dataset [48] includes data on 548 drugs and 48,548 pairwise interactions, along with various types of similarity information pertaining to these drug pairs.

---

[2]`https://github.com/devalab/CIGIN`
[3]`https://github.com/isjakewong/MIRACLE/tree/main/MIRACLE/datachem`

Table 4: Statistics of datasets. $\mathcal{G}^1$ and $\mathcal{G}^2$ are defined in Section 5.1.

| Task | Dataset | | $\mathcal{G}^1$ | $\mathcal{G}^2$ | # $\mathcal{G}^1$ | # $\mathcal{G}^2$ | # Pairs |
|---|---|---|---|---|---|---|---|
| Molecular Interaction | Chromophore [4] | Absorption | Chromophore | Solvent | 6,416 | 725 | 17,276 |
| | | Emission | Chromophore | Solvent | 6,412 | 1,021 | 18,141 |
| | | Lifetime | Chromophore | Solvent | 2,755 | 247 | 6,960 |
| | MNSol [5] | | Solute | Solvent | 372 | 86 | 2,275 |
| | FreeSolv [6] | | Solute | Solvent | 560 | 1 | 560 |
| | CompSol [7] | | Solute | Solvent | 442 | 259 | 3,548 |
| | Abraham [8] | | Solute | Solvent | 1,038 | 122 | 6,091 |
| | CombiSolv [9] | | Solute | Solvent | 1,495 | 326 | 10,145 |
| Drug-Drug Interaction | ZhangDDI [10] | | Small-molecule Drug | Small-molecule Drug | 544 | 544 | 40,255 |
| | ChChMiner [11] | | Small-molecule Drug | Small-molecule Drug | 949 | 949 | 21,082 |

- The **ChChMiner** dataset [49] comprises 1,322 drugs and 48,514 annotated DDIs, sourced from drug labels and scientific literature.

Despite the **ChChMiner** dataset containing a significantly higher number of drug instances compared to the **ZhangDDI** dataset, the number of labeled DDIs is nearly equivalent. This suggests that the **ChChMiner** dataset exhibits a much sparser network of relationships between drugs.

## C   Baselines Setup

To validate the effectiveness of 3DMRL, we primarily evaluate molecular relational learning model architectures trained from scratch for downstream tasks, as well as the same models that are first pre-trained with 3DMRL and then fine-tuned for various downstream tasks. We include the following molecular relational learning model architectures:

- **MPNN** (Message Passing Neural Networks) [11] was originally proposed to predict the various chemical properties of a single molecule. For molecular relational learning tasks, we independently encode each molecule in a pair using MPNN and then concatenate their representations.

  To apply 3DMRL for **MPNN**, we first obtain the atom representation matrices $\mathbf{E}^1$ and $\mathbf{E}^2$ using $f_{2D}^1$ and $f_{2D}^1$, which are MPNNs. Then, we directly use $\mathbf{E}^1$ and $\mathbf{E}^2$ instead of the $\mathbf{H}^1$ and $\mathbf{H}^2$, which considers the interaction between two molecules in Section 3.2. That is, we obtain graph-level embeddings $\mathbf{z}_{2D}^1$ and $\mathbf{z}_{2D}^2$ via $\mathbf{E}^1$ and $\mathbf{E}^2$ with Set2set readout function. Following contrastive learning is done with $\mathbf{z}_{2D}^1$ and $\mathbf{z}_{2D}^2$, and the edge representations $\mathbf{e}_{2D}^{k,l}$ and and initial atom representations for relative geometry $\hat{\mathbf{X}}$ is obtained through $\mathbf{E}^1$ and $\mathbf{E}^2$. One can simply alternate $\mathbf{H}^1$ and $\mathbf{H}^2$ in Section 4 to $\mathbf{E}^1$ and $\mathbf{E}^2$.

- **AttentiveFP** [43] was also initially proposed to predict various chemical properties of individual molecules by employing a graph attention mechanism to gather more information from relevant molecular datasets. For molecular relational learning tasks, we independently encode each molecule in a pair using MPNN and then concatenate their representations.

  More specifically, **AttentiveFP** first obtain atom representation matrices $\mathbf{H}^1$ and $\mathbf{H}^2$ using $f_{2D}^1$ and $f_{2D}^1$, which consist of GAT and GRU layers. Then, the model obtain initial molecule representation $\tilde{\mathbf{z}}_{2D}^1$ and $\tilde{\mathbf{z}}_{2D}^2$ which are further enhanced by considering other molecules in a batch through GAT layers. After passing multiple GAT layers, the model obtain final molecule representations $\tilde{\mathbf{z}}_{2D}^1$ and $\tilde{\mathbf{z}}_{2D}^2$. In our framework, contrastive learning is done with $\mathbf{z}_{2D}^1$ and $\mathbf{z}_{2D}^2$,

---

[4] `https://figshare.com/articles/dataset/DB_for_chromophore/12045567/2`

[5] `https://conservancy.umn.edu/bitstream/handle/11299/213300/MNSolDatabase_v2012.zip?sequence=12&isAllowed=y`

[6] `https://escholarship.org/uc/item/6sd403pz`

[7] `https://aip.scitation.org/doi/suppl/10.1063/1.5000910`

[8] `https://www.sciencedirect.com/science/article/pii/S0378381210003675`

[9] `https://ars.els-cdn.com/content/image/1-s2.0-S1385894721008925-mmc2.xlsx`

[10] `https://github.com/zw9977129/drug-drug-interaction/tree/master/dataset`

[11] `http://snap.stanford.edu/biodata/datasets/10001/10001-ChCh-Miner.html`

and the edge representations $\mathbf{e}_{2D}^{k,l}$ and and initial atom representations for relative geometry $\hat{\mathbf{X}}$ is obtained through $\mathbf{H}^1$ and $\mathbf{H}^2$.

- **CIGIN** (Chemically Interpretable Graph Interaction Network) [32] proposes to model the interaction between the molecules through a dot product between atoms in paired molecules. By doing so, they successfully predict the solubility of drug molecules. We provide detailed descriptions on how to apply 3DMRL for CIGIN in Section 4.

- **CGIB** (Conditional Graph Information Bottleneck) and **CGIB**$_{\text{cont}}$ (Conditional Graph Information Bottleneck with Contrastive Learning)[21] aim to enhance generalization in molecular relational learning by identifying the core substructure of molecules during chemical reactions, based on the information bottleneck theory. While CIGIN is limited to predicting drug solubility, **CGIB** and **CGIB**$_{\text{cont}}$ extend molecular relational learning to predict the optical properties of chromophores in various solvents, molecule solubility in various solvents, and drug-drug interactions.

  **CGIB** and **CGIB**$_{\text{cont}}$ model architectures are highly similar to CIGIN, but they have another branch named *compress module*, which aims to inject noise to the atoms that are not important during the model. Specifically, they obtain $\mathbf{T}^1$ that is node representation matrix with noise, and obtain $\mathbf{z}_{\mathcal{G}_{\text{CIB}}^1}$ from the noise injected matrix along with $\mathbf{z}_{\mathcal{G}^1}$ and $\mathbf{z}_{\mathcal{G}^2}$ which are obtained from $\mathbf{H}^1$ and $\mathbf{H}^2$, respectively. To apply 3DMRL for **CGIB**, we pre-train the model without noise injection module, thereby using $\mathbf{H}^1$, $\mathbf{H}^2$, $\mathbf{z}_{\mathcal{G}^1}$, and $\mathbf{z}_{\mathcal{G}^2}$ in **CGIB** as $\mathbf{H}^1$, $\mathbf{H}^2$, $\mathbf{z}_{2D}^1$, and $\mathbf{z}_{2D}^2$ in Section 4. After pre-training staget, all the modules including noise injection module is trained for the downstream tasks.

In addition to the model architectures, we also compare the recent state-of-the-art molecular pre-training methods based on CIGIN architecture. Since molecular pre-training methods are specifically designed for a single molecule, we pre-train each molecule encoder in CIGIN architecture and adopted the pre-trained weights for molecular relational learning downstream tasks. In Section 5, we include following molecular pre-training approaches:

- **No pre-training** does not involve pertaining process and fine-tune the model using labeled data

- **3D Infomax** [36] increase the mutual information between 2D and 3D molecular representations using contrastive learning

- **GraphMVP** [24] incorporates a generative pre-training framework in addition to contrastive learning

- **MoleculeSDE** [25] designs a denoising framework to capture the 3D geometric distribution of molecules, thereby revealing the relationship between the score function and the molecular force field.

To apply these approaches for MRL, we first pre-train the each encoder $f_{2D}^1$ and $f_{2D}^2$ in Section 3.2 with the above approaches. Then, the pre-trained encoders $f_{2D}^1$ and $f_{2D}^2$ are utilized to output the representations $\mathbf{E}^1$ and $\mathbf{E}^2$, following the remaining pipeline of the model outlined in Section 3.2. That is, each molecule encoder $f_{2D}^1$ and $f_{2D}^2$ implicitly possesses knowledge about the 3D structure of individual molecules, but not the complex interaction geometry between multiple molecules.

## D  Implementation Details

### D.1  Evaluation Protocol

Following Pathak et al. [32], for the molecular interaction prediction task, we evaluate the models under a 5-fold cross-validation scheme. The dataset is randomly split into 5 subsets and one of the subsets is used as the test set, while the remaining subsets are used to train the model. A subset of the test set is selected as the validation set for hyperparameter selection and early stopping. We repeat 5-fold cross-validation three times (i.e., 15 runs in total) and report the accuracy and standard deviation of the repeats. For the DDI prediction task [21], we conduct experiments on two different *out-of-distribution* scenarios, namely **molecule split** and **scaffold split**. For the **molecule split**, the performance is evaluated when the models are presented with new molecules not included in the training dataset. Specifically, let $\mathbb{G}$ denote the total set of molecules in the dataset. Given $\mathbb{G}$, we split $\mathbb{G}$ into $\mathbb{G}_{\text{old}}$ and $\mathbb{G}_{\text{new}}$, so that $\mathbb{G}_{\text{old}}$ contains the set of molecules that have been seen in the training phase, and $\mathbb{G}_{\text{new}}$ contains the set of molecules that have not been seen in the training phase.

Then, the new split of dataset consists of $\mathcal{D}_{\text{train}} = \{(\mathcal{G}^1, \mathcal{G}^2) \in \mathcal{D} | \mathcal{G}^1 \in \mathbb{G}_{\text{old}} \wedge \mathcal{G}^2 \in \mathbb{G}_{\text{old}}\}$ and $\mathcal{D}_{\text{test}} = \{(\mathcal{G}^1, \mathcal{G}^2) \in \mathcal{D} | (\mathcal{G}^1 \in \mathbb{G}_{\text{new}} \wedge \mathcal{G}^2 \in \mathbb{G}_{\text{new}}) \vee (\mathcal{G}^1 \in \mathbb{G}_{\text{new}} \wedge \mathcal{G}^2 \in \mathbb{G}_{\text{old}}) \vee (\mathcal{G}^1 \in \mathbb{G}_{\text{old}} \wedge \mathcal{G}^2 \in \mathbb{G}_{\text{new}})\}$. We use a subset of $\mathcal{D}_{\text{test}}$ as the validation set in inductive setting. In the **scaffold split** setting [14], just like in the molecule split, molecules corresponding to scaffolds that were not seen during training will be used for testing. For both splits, we repeat 5 independent experiments with different random seeds on split data, and report the accuracy and the standard deviation of the repeats. In both scenarios, we split the data into training, validation, and test sets with a ratio of 60/20/20%.

### D.2 Model architecture

For the 2D MRL model, following a previous work [32], we use 3-layer MPNNs [11] as our backbone molecule encoder to learn the representation of solute and solvent for the molecular interaction prediction, while we use a GIN [45] to encode both drugs for the drug-drug interaction prediction task [21]. We utilize a hidden dimension of 56 for molecular interaction tasks and 300 for drug-drug interaction tasks, employing the ReLU activation function for both. For the 3D virtual environment encoder $f_{3D}$, we utilize SchNet [34], which guarantees an *SE(3)-invariant* representation of the environment. For both molecular interaction and drug-drug interaction tasks, we configure SchNet with 128 hidden channels, 128 filters, 6 interaction layers, and a cutoff distance of 5.0.

### D.3 Model training

For model optimization during **Pre-training** stage, we employ the Adam optimizer with an initial learning rate of 0.0005 for the chromophore task, 0.0001 for the solvation free energy task, and 0.0005 for the DDI tasks. The model is optimized over 100 epochs during pre-training.

In the **downstream tasks**, the learning rate was reduced by a factor of $10^{-1}$ after 20 epochs of no improvement in model performance in validation set, following the approach in a previous work [32], with the initial learning rate of 0.005 for the chromophore task, 0.001 for the solvation free energy task, and 0.0005 for the DDI tasks.

**Computational resources.** We perform all pre-training on a 40GB NVIDIA A6000 GPU, whereas all downstream tasks are executed on a 24GB NVIDIA GeForce RTX 3090 GPU.

**Software configuration.** Our model is implemented using Python 3.7, PyTorch 1.9.1, RD-Kit 2020.09.1, and Pytorch-geometric 2.0.3.

## E Additional Experimental Results

### E.1 Molecular Property Prediction Pre-training with Large-Scale Datasets

Although MPP pre-training approaches demonstrate unsatisfactory performance in Section 5, a positive aspect is their ability to leverage large-scale datasets containing both 2D and 3D molecular information. Consequently, we further explore whether utilizing a large-scale pre-training dataset can enhance MPP pre-training strategies in MRL tasks. To do so, we pre-train the encoders with each strategy with randomly sampled 50K molecules in GEOM dataset [2], which consists of 2D topological information and 3D geometric information, following the previous work [24]. In Table 5, we observe that a large-scale pre-training dataset does not consistently result in performance improvements for MRL downstream tasks and can still cause negative transfer in various tasks. On the other hand, we note that MoleculeSTM benefits the most from the large-scale dataset among the strategies, likely due to the complexity of its denoising framework, which necessitates a large-scale dataset to learn the data distribution effectively. Nevertheless, it still exhibits negative transfer in the FreeSolve dataset and performs worse than 3DMRL, highlighting the need for a pre-training strategy specifically tailored to molecular relational learning.

### E.2 Extrapolation in Molecular Interaction Task

The model's generalization ability in out-of-distribution (OOD) datasets is crucial for its application in real-world scientific discovery processes. To this end, we further conduct experiments on molecular interaction tasks by assuming out-of-distribution scenarios, as shown in Table 6. Specifically, we split the dataset based on molecular structure, i.e., molecule split and scaffold split, similar to the approach

Table 5: Performance comparison of CIGIN model on molecular interaction tasks using different pre-training strategies and pre-training dataset (RMSE) (↓). The blue color signifies a positive transfer between the pre-training task and the downstream task, whereas the orange color denotes a negative transfer between the pre-training task and the downstream task. **Pre-training Dataset** indicates the pre-training datasets used during pre-training.

| Strategy | Pre-training Dataset | Chromophore | | | MNSol | FreeSolv | CompSol | Abraham | CombiSolv |
|---|---|---|---|---|---|---|---|---|---|
| | | Absorption | Emission | Lifetime | | | | | |
| No Pre-training | - | 19.66 (0.69) | 25.84 (0.23) | 0.821 (0.017) | 0.567 (0.014) | 0.884 (0.074) | 0.331 (0.029) | 0.412 (0.028) | 0.458 (0.002) |
| **MPP (molecular property prediction) Pre-training** | | | | | | | | | |
| 3D Infomax | MRL | 18.71 (0.61) | 24.59 (0.22) | 0.790 (0.022) | 0.585 (0.015) | 0.873 (0.103) | 0.321 (0.041) | 0.426 (0.036) | 0.464 (0.004) |
| | GEOM | 18.82 (0.24) | 25.14 (0.18) | 0.795 (0.021) | 0.589 (0.027) | 0.899 (0.080) | 0.319 (0.019) | 0.418 (0.023) | 0.466 (0.017) |
| GraphMVP | MRL | 18.40 (0.62) | 24.73 (0.14) | 0.797 (0.022) | 0.561 (0.025) | 1.010 (0.115) | 0.301 (0.025) | 0.418 (0.020) | 0.437 (0.015) |
| | GEOM | 18.85 (0.74) | 24.87 (0.54) | 0.784 (0.014) | 0.551 (0.013) | 0.900 (0.059) | 0.325 (0.007) | 0.410 (0.036) | 0.437 (0.007) |
| MoleculeSDE | MRL | 18.56 (0.24) | 24.91 (0.10) | 0.836 (0.040) | 0.564 (0.018) | 0.971 (0.122) | 0.308 (0.024) | 0.426 (0.028) | 0.454 (0.012) |
| | GEOM | 18.72 (0.16) | 24.77 (0.48) | 0.773 (0.023) | 0.560 (0.086) | 0.909 (0.142) | 0.290 (0.008) | 0.399 (0.034) | 0.449 (0.007) |
| **MRL (molecular relational learning) Pre-training** | | | | | | | | | |
| 3DMRL | MRL | **18.00** (0.17) | **24.21** (0.09) | **0.729** (0.014) | **0.528** (0.019) | **0.839** (0.105) | **0.277** (0.006) | **0.371** (0.031) | **0.435** (0.006) |

used in the DDI task in Section 5. It is important to note that this scenario is significantly more challenging than the out-of-distribution DDI task in Section 5 because it involves a regression task, which can also be viewed as an **extrapolation** task. As shown in Table 6, we observe that pre-training approaches generally benefit model performance in extrapolation tasks, with the exception of one case, namely 3D Infomax for the Lifetime dataset. Among the pre-training approaches, 3DMRL performs the best, underscoring the extrapolation capability of 3DMRL.

Table 6: Performance comparison of the CIGIN model on extrapolation in molecular interaction tasks using different pre-training strategies (RMSE) (↓).

| Strategy | Molecule Split | | | Scaffold Split | | |
|---|---|---|---|---|---|---|
| | Absorption | Emission | Lifetime | Absorption | Emission | Lifetime |
| No Pre-training | 27.51 (0.74) | 37.04 (1.07) | 1.205 (0.033) | 59.55 (1.35) | 60.11 (1.98) | 1.221 (0.033) |
| **MPP (molecular property prediction) Pre-training** | | | | | | |
| 3D Infomax | 27.38 (1.19) | 36.98 (1.24) | 1.257 (0.050) | 58.34 (1.89) | 58.67 (1.00) | 1.207 (0.041) |
| GraphMVP | 26.93 (1.89) | 36.51 (0.92) | 1.201 (0.034) | 59.27 (1.57) | 57.67 (1.14) | 1.199 (0.024) |
| MoleculeSDE | 27.26 (1.19) | 36.48 (1.12) | 1.135 (0.077) | 57.75 (0.74) | 58.74 (1.02) | 1.214 (0.010) |
| **MRL (molecular relational learning) Pre-training** | | | | | | |
| 3DMRL | **25.01** (1.51) | **34.66** (0.89) | **1.033** (0.027) | **57.58** (1.62) | **57.53** (1.13) | **1.178** (0.010) |

### E.3 Ablation Studies

We provide further ablation studies on molecular interaction task and drug-drug interaction task in Table 7 and 8, respectively.

Table 7: Further results from ablation studies on molecular interaction tasks.

| Strategy | Chromophore | | | MNSol | FreeSolv | CompSol | Abraham | CombiSolv |
|---|---|---|---|---|---|---|---|---|
| | Absorption | Emission | Lifetime | | | | | |
| Only Glob. | 18.30 (0.16) | 24.70 (0.16) | 0.739 (0.015) | 0.531 (0.022) | 0.874 (0.060) | 0.301 (0.018) | 0.376 (0.029) | 0.458 (0.014) |
| Only Local | 19.34 (0.50) | 24.80 (0.05) | 0.804 (0.011) | 0.587 (0.019) | 1.184 (0.173) | 0.330 (0.028) | 0.391 (0.020) | 0.466 (0.021) |
| 3DMRL | **18.00** (0.17) | **24.21** (0.09) | **0.729** (0.014) | **0.528** (0.019) | **0.839** (0.105) | **0.277** (0.006) | **0.371** (0.031) | **0.435** (0.006) |

Table 8: Further results from ablation studies on drug-drug interaction tasks.

| Strategy | (a) Molecule Split | | (b) Scaffold Split | |
|---|---|---|---|---|
| | ZhangDDI | ChChMiner | ZhangDDI | ChChMiner |
| Only Glob. | 73.09 (0.83) | 77.68 (0.55) | 73.18 (0.59) | 76.79 (1.13) |
| Only Local | 73.45 (1.29) | 75.93 (1.14) | 73.41 (2.28) | 74.29 (1.79) |
| 3DMRL | **74.00** (0.72) | **78.93** (0.59) | **74.85** (1.58) | **78.56** (1.03) |

### E.4 Further Virtual Interaction Environment Analysis

**Sensitivity Analysis on DDI Datasets.** In Table 9, we provide sensitivity analysis results in drug-drug interaction tasks.

Table 9: Sensitivity analysis on $n$ in drug-drug interaction tasks.

| | Molecule Split | | Scaffold Split | |
|---|---|---|---|---|
| | ZhangDDI | ChChMiner | ZhangDDI | ChChMiner |
| $n = 2$ | 73.77 | 77.15 | 74.76 | 77.01 |
| $n = 5$ | **74.00** | 78.93 | **74.85** | **78.56** |
| $n = 10$ | 73.96 | **79.12** | 74.36 | 77.76 |
| $n = 20$ | 73.94 | 78.75 | 74.03 | 77.64 |

**Further Environment Analysis.** While we propose assigning a single small molecule to each target atom in Section 4.1, we also investigate the impact of varying the number of assigned small molecules per atom in the larger molecule. As illustrated in Figure 6, we observe a decline in model performance as the number of small molecules per atom increases, given a fixed number of target atoms $n$. This suggests that modeling interactions between multiple small molecules and a single atom in a larger molecule can degrade model performance. This is consistent with the scientific understanding that, although hydrogen bonding can occasionally allow multiple molecules to interact with a single atom simultaneously, steric and electronic hindrances frequently impede such interactions. Thus, we contend that our proposed virtual interaction geometry appropriately reflects the real-world physics in molecular interactions.

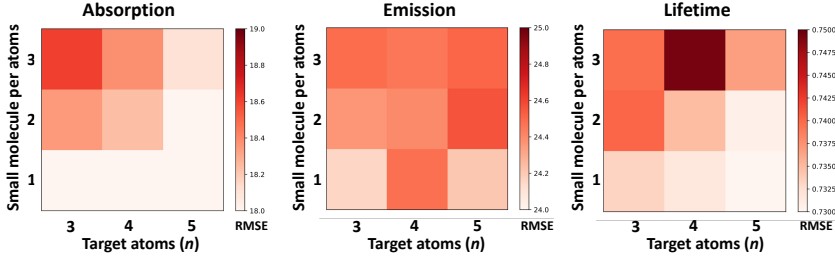

Figure 6: Further environment analysis results.

**Number of Larger Molecules.** In Section 4.1, we initially constructed the virtual geometry in a one-to-many manner (one larger molecule and many smaller molecules) to effectively mimic the

explicit solvent model in traditional MD simulations. However, in this section, we explore the many-to-many configurations between larger molecules and smaller molecules. In Table 10, we observe that the best performance was achieved when there was only one larger molecule. However, since the performance differences were not significant, we can conclude that our model is robust across various configurations.

Table 10: Model performance on different number of larger molecules.

| # Larger molecules | Absorption | Emission | Lifetime |
|---|---|---|---|
| 1 (Ours) | **18.00** | **24.21** | **0.729** |
| 2 | 18.28 | 24.43 | 0.738 |
| 3 | 18.37 | 24.35 | 0.749 |

### E.5  3D Encoder Pre-training Approaches

Since the core concept of our paper is to inject 3D information into a 2D encoder, we choose baseline approaches that pre-train 2D molecular encoder with 3D information. In this section, we compare the approaches that pre-train 3D molecular encoder with 3D information. To do so, since the elaborately calculated 3D structure of the molecules is not available for our datasets, we first calculate the 3D structure of the molecules in the dataset using RDKit ETKDG algorithm. However, some of the molecules in the dataset were not able to obtain 3D structures through RDKit ETKDG algorithm. We excluded these molecules from the experiment, and the results are shown below.

Before Conversion: Absorption – 17,276 pairs, Emission – 18,141 pairs, and Lifetime – 6,960 pairs.

After Conversion: Absorption – 16,756 pairs, Emission – 17,525 pairs, and Lifetime – 6,740 pairs.

Table 11: Performance of various 3D encoder pre-training strategies in RMSE ($\downarrow$). *Note that these results are not directly comparable since some of the molecules in the dataset were not able to obtain 3D structures through RDKit ETKDG algorithm.*

| | Absorption | Emission | Lifetime |
|---|---|---|---|
| 3D-EMGP | 18.62 | **24.06** | 0.753 |
| SliDe | 21.96 | 28.87 | 0.859 |
| Frad | 19.58 | 28.43 | 0.781 |
| 3DMRL | **18.00** | 24.21 | **0.729** |

## F  Pseudocode

In this section, we provide pseudocode of 3DMRL in Algorithm 1.

---

**Algorithm 1** Overall framework of 3DMRL.

---

1: **Input:**
- 2D molecular topology graphs $g_{2D}^1, g_{2D}^2$
- 3D molecular geometric graphs $g_{3D}^1, g_{3D}^2$
- 2D graph encoders $f_{2D}^1, f_{2D}^2$
- 3D Virtual Interaction Geometry Encoder $f_{3D}$

2: **Pre-Training Stage:**
3: **For** epoch **in** epochs:
4:     $\mathbf{z}_{2D}^1, \mathbf{z}_{2D}^2, \mathbf{H}^1, \mathbf{H}^2 = $ 2D MRL ENCODER $(g_{2D}^1, g_{2D}^2)$
5:     $\mathbf{z}_{2D} = (\mathbf{z}_{2D}^1 || \mathbf{z}_{2D}^2)$
6:     $g_{vr} = $ VIRTUAL INTERACTION GEOMETRY CONSTRUCTION $(g_{3D}^1, g_{3D}^2)$
7:     $\mathbf{z}_{3D} = f_{3D}(g_{vr})$                                  /* Virtual Geometry Encoding via SchNet */
8:     $\mathcal{L}_{\text{glob.}} = $ SE(3) INVARIANT GLOBAL GEOMETRY LEARNING $(\mathbf{z}_{2D}, \mathbf{z}_{3D})$
9:     $\mathcal{L}_{\text{local}} = \frac{1}{n} \sum_{i=1}^{n}$ SE(3) EQUIVARIANT LOCAL RELATIVE GEOMETRY LEARNING $(g_{3D}^{2,i}, \mathbf{H}^1, \mathbf{H}^2)$
10:     $\mathcal{L}_{\text{pre-train}} = \mathcal{L}_{\text{glob.}} + \alpha \cdot \mathcal{L}_{\text{local}}$
11:     Update $f_{2D}^1, f_{2D}^2$, and $f_{3D}$
12: **Function** 2D MRL ENCODER $(g_{2D}^1, g_{2D}^2)$
13:     $\mathbf{E}^1 = f_{2D}^1(g_{2D}^1),\quad \mathbf{E}^1 = f_{2D}^2(g_{2D}^2)$
14:     $\mathbf{I}_{ij} = \mathsf{sim}(\mathbf{E}_i^1, \mathbf{E}_j^2)$
        where $\mathsf{sim}(\cdot, \cdot)$ is cosine similarity
15:     $\tilde{\mathbf{E}}^1 = \mathbf{I} \cdot \mathbf{E}^2,\quad \tilde{\mathbf{E}}^2 = \mathbf{I}^\top \cdot \mathbf{E}^1$
16:     $\mathbf{H}^1 = (\mathbf{E}^1 || \tilde{\mathbf{E}}^1),\quad \mathbf{H}^2 = (\mathbf{E}^2 || \tilde{\mathbf{E}}^2)$
17:     $\mathbf{z}_{2D}^1 = \text{Set2set}(\mathbf{H}^1),\quad \mathbf{z}_{2D}^2 = \text{Set2set}(\mathbf{H}^2)$
18:     **return** $\mathbf{z}_{2D}^1, \mathbf{z}_{2D}^2, \mathbf{H}^1, \mathbf{H}^2$
19: **Function** VIRTUAL INTERACTION GEOMETRY CONSTRUCTION $(g_{3D}^1, g_{3D}^2)$
20:     Randomly select $n$ atoms in larger molecule $g_{3D}^1$
21:     Copy small molecule $g_{3D}^2$ to $n$ small molecules $g_{3D}^{2,1}, \ldots, g_{3D}^{2,i}, \ldots, g_{3D}^{2,n}$
22:     Generate a normalized random Gaussian noise vector $\varepsilon$
23:     Create new 3D coordinates for each smaller molecule $g_{3D}^{2,i}$
        $\mathbf{R}^{2,i} = \mathbf{R}^2 + \varepsilon_i * r^2 + \mathbf{R}_i^1$                     /* Broadcasting operation */
24:     Create virtual interaction geometry $g_{vr}$
        $\mathbf{R}_{vr} = (\mathbf{R}^1 || \mathbf{R}^{2,1} || \ldots || \mathbf{R}^{2,i} || \ldots || \mathbf{R}^{2,n})$
        $\mathbf{X}_{vr} = (\mathbf{X}^1 || \mathbf{X}^2 || \ldots || \mathbf{X}^2)$
        $g_{vr} = (\mathbf{X}_{vr}, \mathbf{R}_{vr})$
25:     **return** $g_{vr}$
26: **Function** SE(3) INVARIANT GLOBAL GEOMETRY LEARNING $(\mathbf{z}_{2D}, \mathbf{z}_{3D})$
27:     **return** $\mathcal{L}_{\text{glob}} = -\frac{1}{N_{\text{batch}}} \sum_{i=1}^{N_{\text{batch}}} \left[ \log \frac{e^{\mathsf{sim}(\mathbf{z}_{2D,i}, \mathbf{z}_{3D,i})/\tau}}{\sum_{k=1}^{N_{\text{batch}}} e^{\mathsf{sim}(\mathbf{z}_{2D,i}, \mathbf{z}_{3D,k})/\tau}} + \log \frac{e^{\mathsf{sim}(\mathbf{z}_{3D,i}, \mathbf{z}_{2D,i})/\tau}}{\sum_{k=1}^{N_{\text{batch}}} e^{\mathsf{sim}(\mathbf{z}_{3D,i}, \mathbf{z}_{2D,k})/\tau}} \right]$
28: **Function** SE(3) EQUIVARIANT LOCAL RELATIVE GEOMETRY LEARNING $(g_{3D}^{2,i}, \mathbf{H}^1, \mathbf{H}^2)$
29:     **For** all edges $(k, l)$ **in** $g_{3D}^{2,i}$:
30:     $\mathcal{F}_{k,l} = \left( \frac{\mathbf{r}_k - \mathbf{r}_l}{||\mathbf{r}_k - \mathbf{r}_l||}, \frac{\mathbf{r}_k \times \mathbf{r}_l}{||\mathbf{r}_k \times \mathbf{r}_l||}, \frac{\mathbf{r}_k - \mathbf{r}_l}{||\mathbf{r}_k - \mathbf{r}_l||} \times \frac{\mathbf{r}_k \times \mathbf{r}_l}{||\mathbf{r}_k \times \mathbf{r}_l||} \right),$          /* Construct Orthogonal Frame */
        where $\mathbf{r}_k \in \mathbb{R}^3$ indicates the position of atoms $k$.
31:     $\mathbf{e}_{3D}^{k,l} = \text{Projection}_{\mathcal{F}_{k,l}}(\mathbf{r}_k, \mathbf{r}_l)$                     /* Convert to $SE(3)$-Invariant Feature */
32:     $\mathbf{e}_{2D}^{k,l} = \text{MLP}(\mathbf{H}_k^2 || \mathbf{H}_l^2)$
33:     $\mathbf{e}_{k,l} = \mathbf{e}_{2D}^{k,l} + \mathbf{e}_{3D}^{k,l}$.
34:     $\tilde{\mathbf{X}} = (\mathbf{H}^2 || \mathbf{H}_i^1)$                                         /* Broadcasting operation */
35:     $\mathbf{h}_{k,l} = \text{GNN}(\tilde{\mathbf{X}}, \mathcal{E})$, where $\mathcal{E}$ indicates all edges in $g_{3D}^{2,i}$          /* Obtain Edge Features */
36:     $\hat{f}_k = \sum_l \mathbf{h}_{k,l} \odot \mathcal{F}_{k,l}$                                 /* Convert to $SE(3)$-equivariant Feature */
37:     **return** $\mathcal{L}_{\text{local}} = \frac{1}{N^2} \sum_{k=1}^{N^2} ||f_k^i - \hat{f}_k^i||_2^2$

---

