# OpenReview forum: "3D Interaction Geometric Pre-training for Molecular Relational Learning"
_NeurIPS.cc/2025/Conference — NeurIPS 2025 spotlight_

### Official Review · Reviewer_x5n8 · 2025-07-01

**Clarity:** 4
**Significance:** 4
**Originality:** 3
**Rating:** 5
**Confidence:** 4

**Summary:**

The paper introduces 3DMRL, a novel pre-training framework for Molecular Relational Learning (MRL) that incorporates 3D geometric information into 2D models. It constructs a virtual interaction environment to simulate molecular interactions and uses two pre-training strategies: global geometry learning and local relative geometry learning. The approach significantly improves model performance, with up to 24.93% improvement in various molecular interaction tasks.

**Questions:**

See Weaknesses above.

**Ethical Concerns:**

["NO or VERY MINOR ethics concerns only"]

**Final Justification:**

The paper proposes a novel and effective method for molecular representation learning by incorporating 3D geometric information through a virtual interaction environment, demonstrating strong performance across multiple tasks with comprehensive experiments. The authors addressed concerns regarding model design and validity in the rebuttal, further strengthening the empirical foundation. While the theoretical justification and in-depth discussion on 3D geometric modeling remain limited, the significant experimental results and methodological innovation outweigh this weakness. The paper is well-written and clearly presented. I recommend **acceptance**, with a suggestion to expand the discussion on design rationale and the implications of the geometric modeling in the final version.

**Limitations:**

No. The authors did not demonstrate the limitations of their work.

**Paper Formatting Concerns:**

Nope

**Quality:**

3

**Strengths And Weaknesses:**

# Strengths

1. Novel Contribution: This paper introduces an innovating method for incorporating 3D geometric information into MRL. And the concept of constructing a virtual interaction environment to simulate molecular interactions is an effective and interesting solution to overcome the high computational costs of traditional methods.

2. Effective and rational framework: The use of two complementary pretraining strategies is a good approach for improving the representation of molecular interactions at both macro and micro levels.

3. Comprehensive experimental validation: Extensive experiments across multiple real-world datasets and tasks, which demonstrate that the method consistently improves performance. And this manuscript also covers ablation study, efficiency study, and so on.

4. This manuscript is well-written and logically organized.

# Weaknesses

In my opinion, this manuscript has no significant flaws. The few shortcomings are: this manuscripts lacks of discussions of the proposed approach and theoretical justification.

---

> ### Author Rebuttal · Authors · 2025-07-31
>
> Thank you for strongly championing our work! We are happy to further discuss the justification and other possible views for our method.
>
> ---
> ## Justification for Random Placement of Small Molecules
>
> Randomly placing solvent molecules around a solute is a well-established strategy in molecular simulations. For instance, protein–ligand docking protocols (e.g., Rosetta) often initialize ligands in random orientations relative to the protein before searching for binding modes. Similarly, Monte Carlo insertion methods like Widom’s test-particle approach randomly insert solvent molecules to explore solute–solvent configurations without bias. This stochastic initialization allows efficient sampling over a wide range of distances and orientations while remaining physically sound: in the limit of sufficient sampling, no unphysical configuration is favored, and the process mimics the early stages of solvation when solvent molecules approach from arbitrary directions.
>
> Our approach constructs a “virtual interaction environment” by generating many such random solute–solvent geometries. This approximates the diversity of encounters seen in explicit solvation, while avoiding the high cost of molecular dynamics. It also exposes the model to both favorable and unfavorable orientations, encouraging generalization across interaction types. The method is computationally efficient and theoretically grounded, supported by literature precedent.
>
> ---
> ## Local Relative Geometry as an Interaction Force
>
> While we define local relative geometry for learning fine-grained interactions between molecules, we can view local relative geometry as an interaction force between molecules.
> More specifically, we define a local relative geometry using the normalized vector between a solute atom and its closest solvent atom.
>
> This provides a physically motivated supervision signal rooted in classical intermolecular forces, many of which are central and act along the internuclear axis. For example, van der Waals interactions (described by the Lennard-Jones potential) exhibit repulsive or attractive forces directed along this axis. At short distances, repulsion dominates and aligns directly outward between nuclei.
>
> This supervision scheme serves as a central-force approximation, consistent with classical force fields, and offers a lightweight surrogate for full force labels, which would require costly quantum chemistry or MD simulations. Notably, SchNet [1] demonstrated that even approximate force signals improve learning of molecular interactions. Our direction-based supervision enables the model to learn geometric features like hydrogen bond alignment or steric repulsion trajectories in an SE(3)-equivariant manner.
>
> Since solvent atoms are placed near specific solute atoms, the dominant interaction direction aligns with the interatomic vector, making it a reasonable proxy for the net force axis. Thus, the unit direction vector serves as a pseudo-force label, conveying the primary interaction axis and encouraging the model to encode directionality of intermolecular interactions.
>
> [1] Schütt, Kristof, et al. "Schnet: A continuous-filter convolutional neural network for modeling quantum interactions." Advances in neural information processing systems 30 (2017).
>
> ---
> ## Limitations
>
> Although the limitations of our work are addressed in Section 5.3, we will add a separate section in the camera-ready version to specifically detail concerns regarding the physicality of our virtual environment

---

> > ### Comment · Reviewer_x5n8 · 2025-08-06
> >
> > Thanks for your response. And I will maintain my positive rating.

---

> > > ### Author Response · Authors · 2025-08-06
> > >
> > > Thank you for your response and championing our work!
> > >
> > > We greatly appreciate your valuable feedback and will use our discussion to further improve the manuscript for the final version.

---

### Official Review · Reviewer_8DvQ · 2025-07-02

**Clarity:** 3
**Significance:** 3
**Originality:** 3
**Rating:** 4
**Confidence:** 2

**Summary:**

The paper introduces 3DMRL, a novel 3D geometric pre-training strategy for Molecular Relational Learning (MRL). Traditional MRL approaches are limited to 2D topological structures due to the high cost of acquiring 3D interaction geometry. 3DMRL overcomes this by constructing a 3D virtual interaction environment, allowing 2D MRL models to learn global and local 3D geometric information without expensive quantum mechanical calculations.

**Questions:**

1. Incorporating molecular dynamics trajectories to simulate dynamic interactions beyond static configurations may be virtual environment realism.

**Ethical Concerns:**

["NO or VERY MINOR ethics concerns only"]

**Final Justification:**

I retained my score.

**Limitations:**

Yes, In Section 5.3.

**Paper Formatting Concerns:**

None.

**Quality:**

3

**Strengths And Weaknesses:**

Strengths:

Proposes a cost-effective 3D virtual environment simulating real molecular interactions by arranging small molecules around a larger molecule, replacing expensive quantum calculations.

Dual Pre-training Strategies: SE(3)-Invariant Global Learning and SE(3)-Equivariant Local Learning.

Extensive experiments (15 runs per model), ablation studies, and sensitivity analyses validate the approach's robustness and generalizability.

Weakness:

Relies on RDKit-generated 3D conformations, which may introduce errors for molecules without experimental structures.

Lacks formal mathematical proof of SE(3) equivariance's necessity, relying primarily on empirical results.

---

> ### Author Rebuttal · Authors · 2025-07-31
>
> Thank you for acknowledging our main contributions to the paper: replacing expensive quantum calculations with cost-effective 3D virtual environment simulation!
>
> ---
> ## W1
>
> We agree with the reviewer that relying on ETKDG-generated conformers, while computationally efficient, does introduce some inaccuracy. As you rightly noted, using experimental structures could indeed reduce this error; however, their acquisition comes with substantial expense and often limits the scalability of such methods.
>
> Obtaining experimental 3D structures for molecules, especially for a large and diverse set, is a significant challenge. Techniques like X-ray crystallography or NMR spectroscopy are highly resource-intensive, requiring specialized equipment, significant time, and often large quantities of purified samples. Many molecules are difficult to crystallize or are unstable in conditions required for these experiments. This makes it impractical to generate experimental 3D structures for the vast number of molecules needed for large-scale machine learning model training and evaluation.
>
> A core strength of our work, as you pointed out, is our aim to replace costly operations in traditional QM approaches with AI, thereby reducing overall experimental costs. Introducing the requirement for expensive experimental structures would, unfortunately, create another bottleneck, thereby limiting the practical applicability of our approach to a broader range of molecules and scenarios.
>
> Nevertheless, we concur that incorporating experimental structures, despite the cost and difficulty, could further enhance our model's accuracy. This remains a valuable direction for future work, particularly in contexts where such high-fidelity data is available.
>
> ---
> ## W2
>
> We appreciate the reviewer's comment regarding the perceived lack of a formal mathematical proof for SE(3) equivariance's necessity. We want to clarify that the necessity of SE(3) equivariance in molecular modeling isn't about deriving a mathematical theorem, but rather about aligning our models with a fundamental and inherent property of the physical world.
>
> Specifically, molecules exist in 3D space, and their intrinsic characteristics—such as energy, bond lengths, or interaction forces—are fundamentally independent of their arbitrary position or orientation. If we rotate or translate a molecule in space, its physical nature remains unchanged. For a computational model to accurately capture these true molecular properties, its internal representations and final predictions must transform predictably (or remain invariant, for scalar properties) when the input molecule undergoes such rigid transformations. This is precisely what SE(3) equivariance mathematically enforces and represents, as demonstrated in many previous works [1, 2].
>
> [1] UNKE, Oliver, et al. SE (3)-equivariant prediction of molecular wavefunctions and electronic densities. NeurIPS 2021.
>
> [2] HOOGEBOOM, Emiel, et al. Equivariant diffusion for molecule generation in 3d. ICML 2022.
>
> ---
> ## Q1
>
> Yes, we fully agree that incorporating molecular dynamics (MD) trajectories into our pre-training data would significantly enhance the realism of our virtual environments and further improve our work. Beyond just enriching the environment, this would also open avenues for new pre-training strategies, such as predicting molecular movement rather than just relative geometry.
>
> While acknowledging that calculating extensive MD trajectories introduces a computational bottleneck (as we discussed in our response to W1), our immediate research interest is indeed focused on integrating this valuable MD data into our virtual environment framework.
> We are truly grateful for your insightful suggestion, which highlights a crucial direction for further enhancing our model's realism and capabilities.

---

> > ### Comment · Reviewer_8DvQ · 2025-08-04
> >
> > Thanks for the rebuttal. Thanks for the answers to the molecular dynamics (MD) trajectories.

---

> > > ### Author Response · Authors · 2025-08-06
> > >
> > > We're glad our response, especially regarding the MD trajectories, was helpful!
> > >
> > > We appreciate your valuable feedback and will use our discussion to further improve the manuscript for the final version.

---

### Official Review · Reviewer_1qFd · 2025-07-03

**Clarity:** 3
**Significance:** 3
**Originality:** 3
**Rating:** 5
**Confidence:** 3

**Summary:**

This paper proposes a training strategy for molecular relational learning that involves a 2D and a 3D molecule encoder, where the interaction geometry of two molecules is provided with a computationally efficient approximation to enhance training with notions of the 3D space that molecules live in, while avoiding computationally expensive simulations to obtain the data. The work presents improved performance over baselines, ablation studies, and sensitivity analyses.

**Questions:**

- The rationale for the local geometry learning is not entirely clear to me. If I understand correctly, the target relative geometry $f^i_k$ was constructed randomly in the first place (compare definitions of $R^2$ and $R^1$). Aren’t you learning random noise here?
- What is the added computational expense for constructing the environment and how does it scale with the involved parameters?
- Would you generally recommend to set the number of small molecules $n$ to 5? Does that depend on factors, such as the size of the molecules?

The following questions are speculative and more out of curiosity, I don’t require them to be answered:

- Is this method intended for solute-solvent interactions only or do you expect it to be applicable to eg molecules of similar size or more than two interaction partners?
- Do you expect your method to scale to large biomolecules such as proteins? (both conceptually and computationally)
- I am curious whether the random virtual environment might already provide sufficient information to model real-world interactions (as you argue in 4.1) and what the performance delta to an expensive QM method would be. How do you think (qualitatively) would your method using the virtual environment compare to using QM-derived data?

**Ethical Concerns:**

["NO or VERY MINOR ethics concerns only"]

**Final Justification:**

I am keeping my positive score, although the author discussion was more confusing than clarifying. I was under the impression that the algorithm can be used without ground truth data obtained from conformer generators like ETKDG and use only a simple synthetic interaction environment. This reduces the value of the contribution a bit in my eyes. The more important it is to then please provide quantification of how much complexity your algorithm adds compared to others, either with explicit runtime comparisons or with a complexity statement (Big O). Also, there was no mention of denoising in the method description. If your model is phrased within a denoising framework, please mention that in the revision. I also recommend to clarify the use of the word "geometry" as it seems to be used in multiple ways.

**Limitations:**

The authors discuss a limitation where molecules are placed on top of each other. I would recommend to also touch on the "physicality" of the virtual environment and the limitations of this approximation, as well as computational complexity of its construction (including scaling with various parameters).

**Paper Formatting Concerns:**

Minor: In L360 please reference the equation not the line. I would also consider to use the word “translation” instead of “transition”. The barchart in Fig 3 could use error bars.

**Quality:**

3

**Strengths And Weaknesses:**

This is a very solid contribution to a relevant problem with (mostly) clear description and notation, convincing experimental setups, and good improvement over existing baselines. I especially appreciated the given rationale of the virtual environment in 4.1, the ablation studies, and the sensitivity analysis. I have a few clarifying questions, but would recommend to accept as this work provides an easily applicable improvement that equips molecular prediction models with general 3D geometric information.

---

> ### Author Rebuttal · Authors · 2025-07-31
>
> We appreciate your strong endorsement of our research! It's rewarding to know that our virtual interaction environment construction, ablation studies, and sensitivity analyses resonated positively with you.
>
> ---
> ## Q1
>
> As detailed in Lines 189-191, our approach generates the target geometry anew in each epoch. Crucially, we utilize random Gaussian noise directly without any training on it. While learning this random noise could indeed be a promising research direction, it presents a significant challenge: we lack the ground truth geometry that would typically be obtained through methods like MD simulation. Without such a signal, the noise could easily degenerate to a zero vector, as our model is designed to predict this noise, making a zero vector an "easy target" for the local geometry learning described in Section 4.2.2.
>
> However, if we could obtain MD trajectories of molecular interactions, we could first train a virtual interaction geometry generator to mimic these trajectories. This generator could then be used during our pre-training strategy instead of purely random generation, providing a more informed signal. We're truly grateful for your insightful question; it highlights a crucial direction for further enhancing our model's realism and capabilities.
>
> ---
> ## Q2
>
> As described in our answer to W1, we do not involve any training of noise during environment construction. This key feature means it scales effectively with all involved parameters, irrespective of their quantity.
>
> While pre-training can encounter scaling issues as molecule and environment sizes increase, this concern is isolated to the training phase, not the environment construction itself.
>
> ---
> ## Q3
>
> Yes, we generally recommend setting the number of small molecules, $n$, to 5. As illustrated in Figure 4, we found that this value strikes an optimal balance between computational cost and model performance.
>
> You're right that the optimal $n$ could depend on factors like the size of the molecules. We could dynamically control $n$ by calculating the size ratio between the larger and smaller molecules. However, doing so would introduce an additional hyperparameter that users would then need to tune, potentially adding complexity to the model's application.
>
> However, we believe that exploring how to dynamically learn the environment based on experimental conditions or specific solvent types would be a very promising direction for future work. This would allow our model to adapt to a wider range of realistic scenarios, moving beyond static configurations to more accurately capture the complexities of molecular interactions in diverse experimental settings.
>
> ---
> ## Q4
>
> As detailed in Section 5.1, our model accounts for diverse interaction types, including solute-solvent, chromophore-solvent, and drug-drug interactions. Notably, 3DMRL also handles drug-drug interactions, where interacting molecules are of similar size. This suggests our method is well-suited for modeling interactions between molecules of comparable dimensions.
>
> Currently, our experiments are limited to interactions between two partners. While we believe extending our virtual environment to accommodate more than two distinct interacting molecules (by arranging them without overlap) is straightforward, our 2D encoder parts primarily model pairwise interactions using cross-attention. To effectively model higher-order interactions involving more than two types of molecules, we would need to develop new methods.
>
> Thank you for suggesting us highly promising future work, which is crucial for a deeper understanding of complex molecular systems!
>
> ---
> ## Q5
>
> We believe our method can conceptually be extended to large biomolecules like proteins.
> However, scaling to such large molecules presents practical challenges. Since proteins comprise a significantly greater number of atoms, the resulting virtual environments would become prohibitively large. To address this, we would need to develop specialized methods for constructing virtual environments using larger components than individual atoms, such as amino acid residues. This would be a crucial step for applying our approach to proteins and other large biomolecules.
>
> ---
> ## Q6
>
> Your question regarding the sufficiency of random virtual environments for modeling real-world interactions, and the performance delta compared to expensive QM methods, is insightful. As we discuss in Section 4.1, our goal was to mimic real-world molecular interactions through our virtual environment. The results in Table 3 demonstrate that our virtual environment strikes an effective balance between computational cost and the realism of the generated environment.
>
> On the other hand, we expect that integrating our method with QM-derived data would lead to significant performance improvements. For instance, instead of purely random generation, we could use MD snapshots as initial configurations for our virtual environments, providing a more physically accurate starting point. Furthermore, leveraging entire MD trajectories as pre-training data would allow our model to learn from much richer and dynamic interaction landscapes. While these QM/MD-driven approaches inherently involve a higher computational cost, we believe they offer a path to even more realistically model interaction environments, thereby substantially enhancing our method's predictive power.
>
> ---
> ## Limitations
>
> Thank you for suggesting additional limitations of our work. We will make sure to incorporate the 1) physicality of the virtual environment and 2) computational complexity as additional limitations.
>
> ---
> ## Paper Formatting Concerns
>
> Thank you for your detailed suggestions! We will ensure that all suggestions are included in our camera-ready version.

---

> > ### Comment · Reviewer_1qFd · 2025-08-04
> >
> > I thank the authors for attempting to answer my questions, I would however like to clarify Q1. I am referring to the definition of the loss function of the Local Relative Geometry Learning in Eq. 5. Here, the loss function depends on the relative geometry $f^i_k$ , which in turn is defined in L255 through $R^{2,i}$, which in turn is defined in L170 through a random Gaussian noise vector $\epsilon$ (L166). If I didn't miss anything here, your loss of Eq. 5 therefore attempts to predict something that is at least corrupted by random noise. Why would you do that? Also, as you mention in your answer, you don't have access to ground truth geometry, you refer to it however in your paper in L252. How is that to be understood?

---

> ### Author Response · Authors · 2025-08-05
> **Thank you for asking further clarification**
>
> Thank you for asking for this clarification. We apologize for any confusion our previous response may have caused.
>
> **Regarding Random Gaussian Vector**
>
> As the reviewer correctly pointed out, we construct our virtual environment with random Gaussian noise, i.e., $R^{2, I}$, which is also used to generate $\hat{f}$ in eq (4). However, the original geometry of larger molecule is **ground truth** geometry obtained from RDKit ETKDG algorithm without any added noise.
> Therefore, in Eq (5), our model learns to predict the optimal direction for the smaller molecules to move, simultaneously learning the geometry of larger molecules.
>
> As reviewer mentioned, $f_{k}^{i}$ contains the $R^{2, i}$.
> However, the crucial point is that the model input is also constructed based on the same random noise. Therefore, our model is not predicting something random. Instead, it is learning the deterministic relationship between the noisy input and the corresponding noisy label, which is the geometry of larger molecules. This task is essentially a form of denoising, where the model learns to identify and predict the underlying ground truth structure by mapping a noisy input to its target.
>
> *Denoising in Machine Learning.* Note that adding noise and reconstructing it is widely used in various machine learning field. For example, Diffusion models follow a 'forward process' where noise is gradually added to data to transform it into a simple Gaussian noise distribution. They then learn a 'reverse process' to denoise and restore the original data, thereby learning the complex data distribution. This process forces the model to understand the essential structure of the data from a noisy input and learn how to remove that noise.
>
> **Regarding ground truth geometry**
>
> Moreover, yes, we do not have access to ground truth geometry of whole interaction. That is, we only know the ground truth geometry of each individual molecule, but do not know how they are positioned in the space during the interaction. And in L252, we mentioned it as “ground truth **relative** geometry”, which is defined in L253-255. What we meant here was ground truth of **relative** geometry rather than the whole interaction geometry.
>
> Thank you again for championing our work and for helping us clarify this important point. We will make sure to update the paper to explain this concept more clearly, and happy to discuss further if you have remaining questions!

---

> > ### Comment · Reviewer_1qFd · 2025-08-06
> >
> > As this point will not affect my rating I will close the discussion here, but I recommend to clarify the section 4.2.2 "SE(3)-Equivariant Local Relative Geometry Learning". I thank the authors for the discussion.

---

> > > ### Author Response · Authors · 2025-08-06
> > >
> > > Thank you for your response and for your valuable feedback!
> > >
> > > We'll be sure to improve the clarity of Section 4.2.2 in the final version by incorporating the points from our discussion during the rebuttal.

---

### Official Review · Reviewer_6exS · 2025-07-04

**Clarity:** 3
**Significance:** 3
**Originality:** 3
**Rating:** 4
**Confidence:** 3

**Summary:**

This paper proposes 3DMRL, a pre-training framework that teaches 2 D molecular-relational models to encode 3 D interaction geometry by (a) constructing a randomized “virtual interaction environment” and (b) optimizing global contrastive and local SE(3)-equivariant losses. The expriments show consistent improvements (up to ≈25 % RMSE reduction) over five MRL backbones on eight datasets, plus better OOD generalization than prior 3 D pre-training methods.

**Questions:**

1. How often do generated configurations violate typical inter-molecular distance/angle distributions seen in MD trajectories?
2. As mentioned above about the efficiency concern, can the authors provide the efficiency experiment?

**Ethical Concerns:**

["NO or VERY MINOR ethics concerns only"]

**Final Justification:**

I would recommend this paper as the authors addressed my questions.

**Limitations:**

Yes

**Quality:**

3

**Strengths And Weaknesses:**

Strength
1. The expriments are thorough on 40 downstream tasks with vairous ablation and OOD tests.
2. Method pipeline and algorithm pseudocode are easy to follow, the methods are rational.
3. The proposed component virtual interaction geometry is interesting.
4. The paper is well-written, with sound motivaitons.

Weakness
1. Virtual environment lacks physical validation, where random Gaussian placement may create unrealistic interactions despite collision checks.
2. Comparisons against 3 D encoders rely on ETKDG-generated conformers, weakening those baselines.
3. The pre-training strategies are not novel, global contrastive loss similar to Infomax, thus is very incremental.
4. The efficiency of the method might be a concern.

---

> ### Author Rebuttal · Authors · 2025-07-31
>
> Thank you for acknowledging our effort in extensive experiments of 40 downstream tasks and OOD generalization ability of 3DMRL! We're happy to address each of the weaknesses and questions that the reviewer raised.
>
> ---
> ## W1 and Q1
>
> We acknowledge the reviewer's concern that random Gaussian placement, even with collision checks, might lead to unrealistic interactions and that our virtual environment lacks physical validation. As we detailed in Section 5.3, we're aware that some molecule collisions can still occur.
>
> Our approach prioritizes an efficient balance between computational complexity and performance, as demonstrated in Table 3. Regenerating the environment every time a collision occurs would increase computation time by over four times compared to 3DMRL. Furthermore, as shown in Figure 4, computational complexity significantly increases with the number of solvent molecules ($n$). Considering all possible realistic inter-atomic distances between molecules for every generated configuration would incur an even greater, potentially prohibitive, computational cost. Therefore, while we fully agree on the importance of a physically realistic environment, this was a necessary design choice given the computational constraints and the practicality of our proposed framework.
>
> Regarding the question of how often generated configurations violate typical inter-molecular distance/angle distributions seen in MD trajectories, we couldn't quantitatively assess this as our datasets do not include MD trajectories. However, we truly appreciate you suggesting this promising future work. If we could obtain MD trajectories of molecular interactions, we would first train a virtual interaction geometry generator to mimic these trajectories. This generator could then be used in our pre-training strategy instead of purely random generation, providing a more physically informed signal. Your insightful question highlights a crucial direction for further enhancing our model's realism and capabilities.
>
> ---
> ## W2
>
> We agree with the reviewer that relying on ETKDG-generated conformers, while fast, introduces some inaccuracy. As discussed in Section 4.1, this inherent limitation of computationally generated conformers often leads to the underperformance of previous 3D encoder-based models when compared to methods like 3DMRL, as further evidenced in Appendix E.5.
>
> However, it's crucial to recognize that this is a practical challenge faced by most 3D molecular modeling approaches: obtaining experimentally accurate 3D conformers for the larger molecules is prohibitively expensive and often unfeasible. Therefore, any evaluation of 3D encoder-based models resorts to the datasets that consist of small molecules such as MD17 and QM9.
>
> In contrast, our approach explicitly addresses this challenge through its two-step design. While our pre-training leverages the 3D structural information, our fine-tuning step relies primarily on the more robust and readily available 2D molecular topology. This design makes our model inherently less sensitive to the imperfections of computationally generated 3D conformers during the critical downstream task, allowing for more generalizable and practical applications.
>
> ---
> ## W3
>
> We agree with the reviewer that our global contrastive loss shares similarities with Infomax. However, Infomax is designed exclusively for single-molecule property prediction and cannot learn interaction geometry. To overcome this, we developed virtual interaction geometry, an approach that, to our knowledge, is entirely novel. Beyond global contrastive loss, we also introduce novel local relative geometry learning. This aims to capture localized geometry, which is crucial for understanding how molecules interact in diverse environments. We want to emphasize that the true novelty of our work lies in these innovative methods for modeling molecular interactions.
>
> ---
> ## W4 and Q2
>
> We understand your concerns about the efficiency of our method due to its inherent complexity. However, we'd like to clarify that certain aspects of our approach are highly efficient, while others represent known bottlenecks, primarily from components borrowed from prior work.
>
> As described in Lines 189-191, our virtual environment construction is fast because it relies solely on efficient matrix operations for transitions and rotations. This real-time generation capability is a key strength. Furthermore, as shown in Table 3, 3DMRL maintains a similar computational time across various virtual interaction environment construction variants while consistently demonstrating superior performance on downstream tasks.
>
> On the other hand, the pre-training phase does see an increase in complexity as the size of the virtual environment grows. Specifically, Figure 4(a) illustrates that computational time increases with the number of smaller molecules ($n$) attached to the target atoms. This is primarily because the $f_{3D}$​ encoder model, which we utilize, requires more computation as the input size expands.
>
> To further validate this, we performed time complexity analyses on ablated models. Our findings, presented in the table below, show that the "Only Glob." model, which requires 3D encoding of virtual geometry, demands significantly more training time than the "Only Local" model. This indicates that the 3D encoder is indeed the primary computational bottleneck of our approach.
>
> | | 3DMRL (Full Model) | Only Glob. | Only Local |
> | --- | --- | --- | --- |
> | Time (min/epoch) | 5.32 | 4.30 | 2.30 |
>
> In conclusion, our proposed components, such as virtual geometry construction and local geometry learning, do not introduce additional computational cost. The main efficiency challenge arises from the 3D encoder, a component integrated from previous works.

---

> > ### Comment · Reviewer_6exS · 2025-08-05
> >
> > Thanks for the response, I would maintain my score since it's already positive

---

> > > ### Author Response · Authors · 2025-08-06
> > >
> > > Thank you for your response and for maintaining your positive score! We greatly appreciate your valuable feedback and will use our discussion to further improve the manuscript for the final version.

---

### Decision · Program_Chairs · 2025-09-17

**Decision:**

Accept (spotlight)

**Comment:**

This paper proposes a 3D molecular relational learning framework with a novel 3D geometric pretraining strategy by incorporating a 3D virtual interaction environment, which overcomes the limitations traditional costly mechanisms. The loss is SE(3)-equivariant, and the effectiveness is significant, obtaining up to a 24.93% improvement across 40 tasks.

All the reviewers are existed about the paper with positive reviews. They all feel positive about the significance and novelty of the proposed method. The propose method is novel and shows consistent improvements on various settings such as different backbones and datasets. I agree with the reviewers, and consider that most of the concerns are on some technical and clarification issues, which have been well resolved in the rebuttal. I encourage the authors to incorporate all the discussions from the rebuttal into the final revision, which would further improve the paper.